# Estimation of Changes of Forest Structural Attributes at Three Different Spatial Aggregation Levels in Northern California using Multitemporal LiDAR

**Francisco Mauro** [1,*], **Martin Ritchie** [2], **Brian Wing** [2,†], **Bryce Frank** [1] , **Vicente Monleon** [3], **Hailemariam Temesgen** [1] **and Andrew Hudak** [4]

[1]   Forest Engineering Resources and Management, College of Forestry, Oregon State University, 2150 SW Jefferson Way, Corvallis, OR 97331, USA; bryce.frank@oregonstate.edu (B.F.); temesgen.hailemariam@oregonstate.edu (H.T.)

[2]   US Forest Service Pacific Southwest Research Station, 3644 Avtech Parkway, Redding, CA 96002, USA; mritchie@fs.fed.us (M.R.)

[3]   US Forest Service Pacific Northwest Research Station, 3200 SW Jefferson Way, Corvallis, OR 97331, USA; vjmonleon@fs.fed.us

[4]   US Forest Service, Rocky Mountain Research Station, 1221 S Main St, Moscow, ID 83843, USA; ahudak@fs.fed.us

*   Correspondence: francisco.mauro@oregonstate.edu

†   Deceased.

**Abstract:** Accurate estimates of growth and structural changes are key for forest management tasks such as determination of optimal rotation times, optimal rotation times, site indices and for identifying areas experiencing difficulties to regenerate. Estimation of structural changes, especially for biomass, is also key to quantify greenhouse gas (GHG) emissions/sequestration. We compared two different modeling strategies to estimate changes in V, BA and B, at three different spatial aggregation levels using auxiliary information from two light detection and ranging (LiDAR) flights. The study area is Blacks Mountains Experimental Forest, a ponderosa pine dominated forest in Northern California for which two LiDAR acquisitions separated by six years were available. Analyzed strategies consisted of (1) directly modeling the observed changes as a function of the LiDAR auxiliary information ($\delta$-modeling method) and (2) modeling V, BA and B at two different points in time, including a term to account for the temporal correlation, and then computing the changes as the difference between the predicted values of V, BA and B for time two and time one. We analyzed predictions and measures of uncertainty at three different level of aggregation (i.e., pixels, stands or compartments and the entire study area). Results showed that changes were very weakly correlated with the LiDAR auxiliary information. Both modeling alternatives provided similar results with a better performance of the $\delta$-modeling for the entire study area; however, this method also showed some inconsistencies and seemed to be very prone to extrapolation problems. The $y$-modeling method, which seems to be less prone to extrapolation problems, allows obtaining more outputs that are flexible and can outperform the $\delta$-modeling method at the stand level. The weak correlation between changes in structural attributes and LiDAR auxiliary information indicates that pixel-level maps have very large uncertainties and estimation of change clearly requires some degree of spatial aggregation; additionally, in similar environments, it might be necessary to increase the time lapse between LiDAR acquisitions to obtain reliable estimates of change.

**Keywords:** forest structure change; EBLUP; small area estimation; multitemporal LiDAR and stand-level estimates

## 1. Introduction

　　Light detection and ranging LiDAR data have been extensively used in forest inventories to provide auxiliary information that is highly correlated with multiple forest structural attributes [1–3]. This strong correlation allows estimating forest structural attributes more efficiently than if only field measurements are available [4]. In addition, the spatially explicit nature of LiDAR enables the mapping of forest attributes at fine resolutions (e.g., [2,5]). Accurate estimates of growth and structural changes are key for forest management as multiple management tasks such as determination of optimal rotation times, calculation of site indexes or the identification of areas experiencing difficulties in regeneration. Estimation of biomass is also key to quantifying greenhouse gas (GHG) emissions/sequestration, to comply with the International Panel on Climate Change (IPCC) reporting and good practice guidelines [6], and to develop a correct appraisal of forest resources for carbon markets. The extensively used area based approach (ABA) [1] provides a way to estimate forest attributes at multiple levels ranging from single pixels to large areas using LiDAR auxiliary information [7]. Availability of repeated LiDAR data acquisitions has opened the door to estimation of changes in forest structural attributes over time (e.g., [8,9]) using the ABA method.

　　In the ABA, the area under study is covered by a regular grid that will define a population of pixels or grid cells. In this approach, the field plots used to train models and the grid cells are of the same size, typically between 400 m$^2$ and 900 m$^2$. A direct application of predictive models will render predictions for grid units of size too small to be considered of interest for reporting in forest inventories. Areas of interest (AOIs) (i.e., the areas for which estimates are needed) are typically geographic units that can vary in size depending on the particular application. For worldwide inventories or inventories over continents or countries, AOIs are typically administrative or political units such as countries or municipalities. In forest management applications, AOIs are typically stands, compartments or even complete forests or landscapes. All these AOIs require spatial aggregation of grid units. However, validations of predictive models in the ABA literature are typically performed using global metrics of model fit, such as the sample-based root mean square error or bias, that provide average measures of uncertainty for predictions made for pixels or plots. These measures of uncertainty derived from the model fitting stage do not directly translate into measures of uncertainty for predictions for AOIs composed of multiple pixels (i.e., countries, municipalities, forests, stands, etc.). In addition, even when considering single pixels, they are not AOI-specific, as they only provide an average value, across the entire population, of the error that can be expected using a given model.

　　Thus, it is clear that uncertainty measures used as quality controls in forest inventories need to be made at the AOI-level and change estimation is not an exception. For large areas holding large sample sizes, AOI-specific estimates of means or totals and their measures of uncertainty can be obtained using direct estimators (e.g., [10–12]) that use only use sample data from the AOI under consideration. However, if the AOI sample sizes are not large enough to support direct estimates with reliable precision, then they must be regarded as small areas [13].

　　Small area estimation (SAE) techniques, especially empirical best linear unbiased predictors (EBLUPs) in combination with the ABA approach have been used to obtain estimates, and their corresponding measures of uncertainty for subpopulations such as municipalities [14], groups or management units [15] and stands [4,14,16,17]. SAE techniques allow correcting the potential bias problems of synthetic predictions (i.e., predictions developed assuming that a general model developed at the population level holds for all subpopulations) and also permit reducing the large variance problems of direct estimators when AOIs sample sizes are small [13]. In addition, while EBLUPs have been extensively used in SAE contexts, they can also be used to produce estimates for subpopulations or AOIs with large sample sizes and preserve important advantages over other methods. First, they allow obtaining model-unbiased estimates and their corresponding measures of uncertainty for all AOIs using a single model that explicitly considers potential variations between AOIs. This is a clear advantage over synthetic methods that assume that a certain relation derived for the entire population holds in all AOIs. A second advantage of EBLUPs is that it is possible to reduce the modeling effort

required by direct model-based or model-assisted methods where a model is needed for each AOI. It is thus clear that SAE techniques in combination with LiDAR auxiliary information have potential applications in multiple forest inventories contexts. Unfortunately, to the best of our knowledge, all studies on SAE and forest inventories have focused on estimation of structural attributes at a given point in time, and little is known about: (1) their performance when applied to forest structure change estimation, and (2) about how these techniques compare to other methods used for estimation of changes in AOIs comprising entire populations [10,12,18,19] and especially subpopulations [20].

In this study, we analyzed the two most commonly used strategies to model changes in structural forest attributes using repeated LiDAR acquisitions, and analyzed their performance when used to obtain EBLUPs for AOIs of different size. The first strategy, referred hereafter as the $\delta$-modeling method, considers the change, $\delta$, over the time between LiDAR acquisitions as the model response. The second strategy, which we will call $y$-modeling method, focuses on modeling the structural attributes $y$, and their derived change over time. As a novelty, in the $y$-modeling method, the temporal correlation of both model errors and AOI random effects were taken into account. We considered changes in three structural variables, and AOIs at three different spatial aggregation levels in order to provide insights for future applications where estimates for an entire population and for subpopulations of different sizes are needed. Variables under study are standing volume (V), above ground biomass (B) and basal area (BA) and AOIs subject to analysis are (1) an entire forested area or landscape, (2) subpopulations that in this case are forest stands and (3) pixels as gridded maps are common output in mapping applications.

## 2. Materials and Methods

### 2.1. Study Area

The study area is Blacks Mountains Experimental Forest (BMEF), a 3715 ha forest managed by the United States Forest Service, located northeast of Lassen National Park in northern California, USA (Figure 1). Elevation ranges from 1700 m to 2100 m above sea level. Slopes are gentle (<10%) on the lower parts of the forest and moderate (10%–40%) at higher elevations. Climate is Mediterranean with a certain degree of continentality, with dry summers and wet and cold winters when precipitation is in the form of snow. Average precipitation is 460 mm per year with monthly average temperatures that range from −9 °C to 29 °C. Soils are developed over basalts with depths that range from 1 to 3 m. Ponderosa pine (*Pinus ponderosa Lawson* & C. Lawson) dominated forest occupies the majority of the area. Incense cedar (*Calocedrus decurrens* (Torr.) Florin), white fir (*Abies concolor* (Gordon & Glend) Hildebr) and Jeffrey Pine (*Pinus jeffreyi* Grev & Balf.) are abundant accompanying species. Forest structure is relatively open and the canopy cover varies greatly within the forest (see Figure 1). A more detailed description of the study area can be found in [21,22].

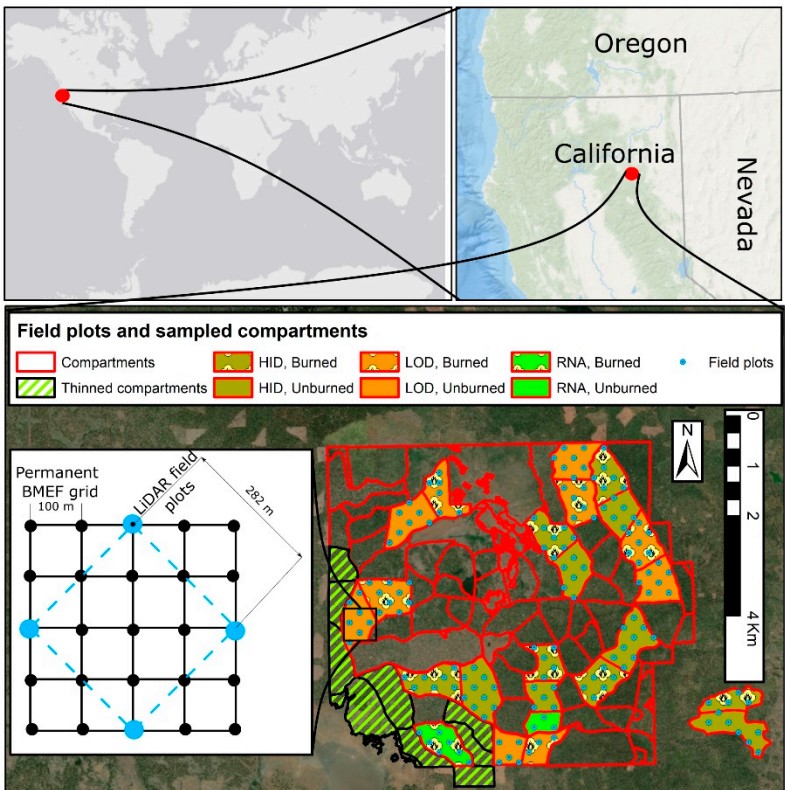

**Figure 1.** Study area location map, delineated stands and field plots, and detailed diagram showing the light detection and ranging LiDAR field plots grid over the permanent Blacks Mountains Experimental Forest (BMEF) grid of permanent makers.

## 2.2. Sampling Design and Field Data

In total, 106 forested stands were delineated in BMEF. Small non-forested patches were masked in the study area and hence were not considered part of the population under study. Out of the 106 forested stands, 24 were selected and sampled in the field. Nine of the remaining 82 unsampled stands were subject to thinning during the period between two available LiDAR acquisitions (i.e., 2009–2015) and all thinning operations were finished by fall 2011. These nine stands are located on the southwestern edge of BMEF and were analyzed separately because the sample of field plots used to train the LiDAR models did not include any stand subject to similar silvicultural interventions (Figure 1). Sampled stands come from a long-term research project initiated in BMEF in 1991 and, excluding the nine thinned stands, were representative of the forest structures and forest management treatments applied in rest of BMEF.

Sampled stands were subject to six different types of treatments resulting from crossing two different factors. The first factor is the structural diversity. It has three levels referred hereafter as low structural diversity (LoD), high structural diversity (HiD) and research natural areas, RNA, or controls. Low structural diversity stands are subject to thinning operations aiming to generate simplified single-strata structures. High diversity stands are subject to thinning where all canopy layers and age groups are preserved, resulting in a multi-storied forest structure with trees of different sizes and ages. Neither the HiD stands nor the LoD stands were subject to thinnings during the period between the two available LiDAR flights. Finally, RNA stands are not subject to any thinning or harvest operation. In total, 10 LoD, 12 HiD and two RNA stands were measured in the field. The second factor under consideration was the presence or absence of prescribed forest fires. Half of the LoD, HiD and RNA stands sampled in the field had been subject to prescribed fires, but only one of the RNA stands was subject to prescribed fires during the period 2009–2015.

A sample of 151, 16 m radius plots (804 m$^2$) were measured in the field during the summer of 2009 and then remeasured during the summer of 2016. All field plots were located on nodes of the 100 m by 100 m grid of monumented markers at BMEF. Coordinates of the makers were determined using traverse methods and survey grade GPS observations and have an accuracy of 15 cm or better (see [23]). For each of the 26 stands selected for sampling, a node of the BMEF 100 m grid was randomly selected and used as a starting node for a 282 m by 282 m grid formed by selecting every other plot of the 100 m grid moving in the diagonal directions. Field measurements were taken on the nodes of the 282 m by 282 m grid (see Figure 1).

Within each field plot all live trees with DBH larger than 9 cm, and all dead standing trees with DBH larger than 12 cm, were stem mapped and measured for DBH and height. Plot basal area (BA) was derived directly from the field measurements. Volume (V) and above ground biomass (B) were computed as the sum of the individual tree volumes and biomasses of all standing trees. Individual tree volumes and biomasses were estimated using species-specific allometric models included in the national volume estimation library (NVEL) and in the national biomass estimation library (NBEL). To account for the one-year difference between acquisition of field measurements in 2016, and the second LiDAR data acquisition obtained in 2015; plot-level values of the variables under analysis were computed for 2015 by linearly interpolating between the values obtained for 2009 and 2016. Finally, for each field plot we computed the change of V, B and BA on a per year basis, as the difference of the plot-level values in 2009 and 2015 divided by 6. For two plots close to the southeastern boundary of the forest, changes in V were extremely large, more than three standard deviations away from the mean value for the change in volume. These anomalous plots were removed from the analysis because such large changes seemed to be derived from edge effects. Plot-level values for 2009, 2016 and per year increments for the period, 2009-2015, for V, B and BA, in the remaining 149 plots are summarized in Table 1.

**Table 1.** Minimum (Min), mean (Mean), standard deviation (Sd), and maximum (Max) of the plot-level values for 2009, 2015 and yearly increments for the period 2009–2015. Values of volume V, basal area BA and biomass B are expressed on a per-hectare basis.

| Variable (Units) | Period | Min | Mean | Sd | Max |
|:---:|:---:|:---:|:---:|:---:|:---:|
| V(m$^3$ ha$^{-1}$) | | 19.87 | 166.93 | 119.66 | 619.43 |
| BA(m$^2$ ha$^{-1}$) | 2009 | 3.81 | 23.43 | 12.02 | 66.54 |
| B(Mg ha$^{-1}$) | | 8.31 | 83.65 | 61.55 | 323.30 |
| V(m$^3$ ha$^{-1}$) | | 17.20 | 175.52 | 117.04 | 644.30 |
| BA(m$^2$ ha$^{-1}$) | 2015 | 3.42 | 25.45 | 12.01 | 67.47 |
| B(Mg ha$^{-1}$) | | 8.34 | 89.38 | 60.29 | 335.03 |
| V(m$^3$ ha$^{-1}$year$^{-1}$) | | −10.89 | 1.43 | 3.88 | 11.19 |
| BA(m$^2$ ha$^{-1}$year$^{-1}$) | Increment 2009–2015 | −0.91 | 0.34 | 0.45 | 1.74 |
| B(Mg ha$^{-1}$year$^{-1}$) | | −5.81 | 0.95 | 1.97 | 5.99 |

For the nine unsampled stands thinned during the period 2009–2015 all thinning operations were completed by fall 2011. In total 427.40 hectares were thinned with prescriptions that varied among stands. Approximately 80% of the area was thinned from below, leaving a residual basal area of 17.22 m$^2$ ha$^{-1}$ to 25.25 m$^2$ ha$^{-1}$. For the remaining 20% of the area, approximately one quarter was not thinned while the other three quarters were thinned to a residual BA that ranged from 6.89 m$^2$ ha$^{-1}$ to 13.77 m$^2$ ha$^{-1}$. Fresh weight of total extractions for the 427.40 hectares subject to thinning was 11,009.38 Mg of logs and 23,164.32 Mg of chipped material.

*2.3. LiDAR Data Acquisitions*

Two LiDAR acquisitions are available for BMEF. The first LiDAR dataset was acquired during the summer of 2009 using a Leica ALS 50 discrete return sensor. Flying altitude was 900 m, side-lap

between adjacent flight lines was at least 50% and scanning angle was ±14°. The LiDAR data vendor generated digital terrain models (DTMs) with an accuracy of 15 cm at 95% confidence level. Additional details on the LiDAR data collection for the 2009 acquisition can be found in [24]. The same vendor in the study area performed a second LiDAR acquisition during the summer of 2015 using the same sensor, flying altitude and side-lap specifications. DTMs were also created for 2015 by the vendor.

Four sets of auxiliary variables were considered in this study. The first two sets are composed of 42 LiDAR predictors computed for each acquisition date. Set 1 will represent the predictors for 2009 and Set 2 the predictors for 2015. These predictors are descriptors of the point cloud height distributions and were all relative quantities to avoid introducing noise due to local differences in the point cloud densities of 2009 and 2015 [25]. The third set of predictors, Set 3, was computed as the differences between the 2009 and the 2015 LiDAR predictors. Finally, the fourth set of predictors, Set 4, included the incoming solar radiation computed using the Environmental Systems Research Institute (ESRI) ArcGIS Area Solar Radiation tool [26] with the 2009 digital surface model (DSM) as input; and two treatments: (1) single- or multi-story structural diversity and (2) presence or absence of prescribed fires. All predictors were computed for each field plot and for a grid with a cell size of 805 m$^2$ covering the entire BMEF. The cell size matched the field plot size and each cell of the grid was considered a population unit, equivalent to the field plots. Predictors and their corresponding acronyms used in further sections are summarized in Table A1.

## 2.4. AOIs, Target Parameter and Overview of Modelling Strategies

Two different types of subsets of population units will be repeatedly used throughout the manuscript in remaining sections. These subsets and their corresponding notation are: the sample of plots measured in the field, denoted using sub-index $s$ and the target AOIs represented by pixels, denoted using sub-index $\alpha$.

Three different groups of AOIs representing different levels of spatial aggregation were analyzed. The first group represents the largest level of spatial aggregation and represents the entire population under study. Within this group, we considered the set of all sampled stands, SS, and the entire BMEF study area after removing the nine thinned and unsampled stands, SA (i.e., sampled and unsampled but not thinned stands). The second group consists of the 106 forested stands in BMEF. In this group, we considered separately the unsampled and thinned stands (nine stands), unsampled and not thinned stands (73 stands) and sampled and not thinned stands (24 stands). Finally, the third group is the set of all pixels of the LiDAR grid covering the forested area in BMEF.

The main objective of this study was to analyze AOI-specific estimates of the change between 2009 and 2015 for three different structural variables, (V, B and BA). We will use the generic term variable of interest and the letter $y$ to refer to the forest structural variables, and the term target parameter and Greek letter $\Delta$ to refer to the quantities that we seek to estimate. Hereafter, target parameters will always refer to changes over time for the totals of the variables of interest in the considered AOIs, and will be expressed in a per hectare and year basis.

Considering that all pixels have the same area, the target parameter $\Delta_\alpha$ for a generic AOI or subset of population units, $\alpha$, can be expressed as:

$$\Delta_\alpha = \sum_{i=1}^{N_\alpha} K_{y\alpha}(y_{i\alpha 15} - y_{i\alpha 09}) = \sum_{i=1}^{N_\alpha} K_{\delta\alpha}\delta_{i\alpha}, \tag{1}$$

where $N_\alpha$ is the number of population units (i.e., pixels) in the AOI. The terms $y_{i\alpha 15}$ and $y_{i\alpha 09}$ respectively represent the value of the variable of interest for 2009 and 2015 for the $i^{th}$ population unit of $\alpha$, and $\delta_{i\alpha}$ is the change, for the $i^{th}$ pixel of $\alpha$, in the variable of interest during the period 2009 to 2015. Finally, for comparability with previous studies, the variables of interest will be expressed in a per unit area basis, and the increments $\delta_{i\alpha}$ will be expressed in a per unit area and year basis. Thus, to ensure that $\Delta_\alpha$ is expressed in the correct units, it is necessary to introduce the factors $K_{y\alpha}$ and $K_{\delta\alpha}$. When $y_{i\alpha 15}$

and $y_{i\alpha 09}$ are expressed in a per unit area basis $K_{y\alpha} = \frac{1}{6N_\alpha}$ and for $\delta_{i\alpha}$ in a per unit area and year basis $K_{\delta\alpha} = \frac{1}{N_\alpha}$.

We calculated AOI estimates using two different methods. The first, $\delta$-modeling method, for estimation of change uses models similar to those in approach A5 of Poudel et al. [8]. In this approach, the change in a structural variable at the plot/pixel-level ($\delta_{i\alpha}$) is directly modeled as a function of the LiDAR auxiliary variables available for the study area. The second, the $y$-modeling method, uses a modified version of approach A4 of Poudel et al. [8] to obtain AOI-specific estimates of change. Models in this approach jointly relate structural variables ($y_{\alpha 15}$ and $y_{\alpha 09}$) and LiDAR auxiliary information at a given point in time and account for the correlation between errors obtained for the same plot/pixel at different times. For both methods, variability between stands was accounted by considering them as small areas. Thus, stand-level random effects were included in the models.

### 2.5. δ-Modeling Method

### 2.5.1. Model δ-modeling Method

Models in the $\delta$-modeling method relate the change (per year) of the variable of interest in a population unit to the auxiliary variables for the population unit. To indicate that these models consider change in the variables of interest directly, model parameters, stand-level random effects and model errors will include the subscript $\delta$. Three different types of auxiliary variables were considered as potential predictors in the $\delta$-modeling method. First, changes in the LiDAR auxiliary variables for the period 2009-2015, Set 3, were considered following Poudel et al. [8] as changes in LiDAR predictors are expected to correlate with growth or changes in forest attributes. Forest structure relates to growth. Thus, the LiDAR auxiliary variables for 2009, Set 1, were also considered as potential predictors that act as proxies for forest structure at the beginning of the period 2009-2015. Finally, the incoming solar radiation and the structural diversity factors and presence of prescribed fires, Set 4, were also considered as potential predictors.

For the $j^{th}$ population unit in the $i^{th}$ stand, models of the $\delta$-modeling method have the form:

$$\delta_{ij} = x_{\delta ij}^t \beta_\delta + v_{\delta i} + \varepsilon_{\delta ij}, \tag{2}$$

where $t$ indicate the transpose operator and $x_{\delta ij}^t$ is a vector of auxiliary variables in which the first element takes the value 1 for the intercept. The term $\beta_\delta$ is a vector of model coefficients where the first element is the intercept of model (2). Selection of auxiliary variables included in the model was performed using the method described in [27]. Stand-level random effects $v_{\delta i}$ are assumed to be independently and identically distributed (i.i.d.) normal random variables $v_{\delta i} \sim N(0, \sigma_{\delta v}^2)$ for all $i = 1, \ldots, D$, where $D$ is the total number of stands in the study area. Model errors are i.i.d. normal random variables $\varepsilon_{\delta ij} \sim N(0, \sigma_{\delta \varepsilon}^2)$ independent of the stand-level random effects (i.e., $Cov(\varepsilon_{\delta ij} v_{\delta,k}) = 0$, for all $i$, $j$ and $k$). Models with spatially correlated errors and with non-constant error variances were initially considered but discarded in the model selection stage, as they were not found to be significant (see Section 2.5.3).

For a generic set of population units denoted by subscript $\xi$ (which can represent either $s$ or $\alpha$), the relation in matrix notation between vector of changes of structural variables $\delta_{y\xi}$, and the auxiliary variables included in the model ($X_{\delta\xi}$), is expressed as:

$$\delta_\xi = X_{\delta\xi} \beta_\delta + Z_{\delta\xi} v_\delta + \varepsilon_{\delta\xi}, \tag{3}$$

where $\delta_\xi = (\delta_1, \ldots, \delta_{N_\xi})^t$, with $\delta_k$ being the yearly change for the forest structural variable $y$, in the $k^{th}$ unit of $\xi$, and $N_\xi$ is the number of elements in the set $\xi$. The $k^{th}$ element of $\xi$ will be an element of a given stand. To explicitly indicate this membership we will use, when necessary, the sub-indexes $i^{th}$ and $j^{th}$ to respectively indicate the stand and index of the element within the stand. The $k^{th}$ row of the matrix $X_{\delta\xi}$ is $x_{\delta k}^t$. The vector $v_\delta = (v_{\delta 1}, \ldots, v_{\delta D})^t$ is a vector of stand-level random effects with

variance covariance matrix $\boldsymbol{G}_\delta = \sigma^2_{\delta v} \boldsymbol{I}_D$, where $\boldsymbol{I}_D$ is the identity matrix of dimension $D$. The matrix $\boldsymbol{Z}_{\xi\delta}$ is a $N_\xi$ x $D$ incidence matrix that describes stand membership for each population unit. The $r^{th}$ row of $\boldsymbol{Z}_{\xi\delta}$ have zeros at all positions except at position $i$, where $i$ is the index of the stand to which the $k^{th}$ unit of $\xi$ belongs. Finally, $\boldsymbol{\varepsilon}_{\xi\delta}$ is a vector of $N_\xi$ model errors with diagonal variance covariance matrix $\boldsymbol{R}_{\xi\delta} = \sigma^2_{\delta\varepsilon} \boldsymbol{I}_{N_\xi}$.

To simplify the notation, hereafter $\boldsymbol{\theta}_\delta = (\sigma^2_{\delta\varepsilon}, \sigma^2_{\delta v})$, will represent the vector of variance parameters. The variance covariance matrix of $\boldsymbol{\delta}_\xi$ is:

$$\boldsymbol{V}_{\delta\xi}(\boldsymbol{\theta}_\delta) = \boldsymbol{Z}_{\delta\xi}\boldsymbol{G}_\delta(\boldsymbol{\theta}_\delta)\boldsymbol{Z}^t_{\delta\xi} + \boldsymbol{R}_{\delta\xi}(\boldsymbol{\theta}_\delta). \tag{4}$$

Model (3) is a linear mixed effect model and a special case of the basic unit-level described in [28] (pp. 174).

### 2.5.2. Target Parameter $\delta$-modeling Method

Under model (2) the target parameter (1) for a generic AOI $\alpha$ can be expressed as:

$$\Delta_\alpha = \frac{1}{N_\alpha}\sum_{i=1}^{N_\alpha} \delta_{i\alpha} = \frac{1}{N_\alpha}\boldsymbol{1}^t\boldsymbol{\delta}_\alpha = \frac{1}{N_\alpha}\boldsymbol{1}^t(\boldsymbol{X}_{\delta\alpha}\boldsymbol{\beta}_\delta + \boldsymbol{Z}^t_{\delta\alpha}\boldsymbol{v}_\delta + \boldsymbol{\varepsilon}_{\delta\alpha}) = \boldsymbol{l}^t_{\delta\alpha}\boldsymbol{\beta}_\delta + \boldsymbol{m}^t_{\delta\alpha}\boldsymbol{v}_\delta + \boldsymbol{q}^t_{\delta\alpha}\boldsymbol{\varepsilon}_{\delta\alpha}, \tag{5}$$

Thus, the target parameter is a linear target parameter similar to the one considered in [4] where $\boldsymbol{1}^t$ is a vector of ones and $\boldsymbol{l}^t_{\delta\alpha} = \frac{1}{N_\alpha}\boldsymbol{1}^t\boldsymbol{X}_{\delta\alpha}$, $\boldsymbol{m}^t_{\delta\alpha} = \frac{1}{N_\alpha}\boldsymbol{1}^t\boldsymbol{Z}^t_{\delta\alpha}$ and $\boldsymbol{q}^t_{\delta\alpha} = \frac{1}{N_\alpha}\boldsymbol{1}^t$ are vectors of known constants for the target AOI $\alpha$.

### 2.5.3. Model Selection and Estimator $\delta$-modeling Method

The target parameter $\Delta_\alpha$ was estimated for all considered AOIs using $\hat{\Delta}_\alpha$ the empirical best linear unbiased predictor (EBLUP) described in [29]. For each variable of interest, auxiliary variables included in $\boldsymbol{X}_{\delta\alpha}$ were preselected using the best subset selection procedure described in [29] (pp. 179–180). When models with similar values of model root mean square error or coefficients of determination were compared, the preferred option was to select the model with smallest values of $\sigma^2_{\delta v}$. This criterion is appropriate to minimize the leading term of the AOI specific mean square errors [29] (pp. 176). Pre-selected models considered constant model error variances and no spatial correlation of model errors were fitted using maximum likelihood (ML). In a subsequent stage, models were re-fitted using ML including: (1) an exponential spatial correlation model for the model errors and (2) a non-constant error variance where $\varepsilon_{\delta ij} \sim N(0, \sigma^2_{\delta\varepsilon} k^{2w_\delta}_{ij})$. The term $k_{ij}$ is the value of the predictor included in the model most correlated to $\delta$ and $w_\delta$ is an additional parameter to account for heteroscedasticity. For all variables, no clear patterns of spatial correlation or non-constant variances were observed, which supports the model form described in Section 2.5.1.

Final estimates $\hat{\boldsymbol{\theta}}_\delta$ of the variance parameters $\boldsymbol{\theta}_\delta$ were obtained using restricted maximum likelihood (REML) with the R [30] package nlme [31]. REML estimates $\hat{\boldsymbol{\beta}}_\delta(\hat{\boldsymbol{\theta}}_\delta)$ of $\boldsymbol{\beta}_\delta$ were functions of the estimated variance parameters (6):

$$\hat{\boldsymbol{\beta}}_\delta(\hat{\boldsymbol{\theta}}_\delta) = \{\boldsymbol{X}^t_{\delta s}\hat{\boldsymbol{V}}_{\delta s}(\hat{\boldsymbol{\theta}}_\delta)^{-1}\boldsymbol{X}_{\delta s}\}^{-1}\boldsymbol{X}^t_{\delta s}\hat{\boldsymbol{V}}_{\delta s}(\hat{\boldsymbol{\theta}}_\delta)^{-1}\boldsymbol{\delta}_s. \tag{6}$$

Matrices $\hat{\boldsymbol{V}}_{\delta s}(\hat{\boldsymbol{\theta}}_\delta)$, $\hat{\boldsymbol{G}}_\delta(\hat{\boldsymbol{\theta}}_\delta)$ and $\hat{\boldsymbol{R}}_{\delta s}(\hat{\boldsymbol{\theta}}_\delta)$ are obtained replacing the estimated variance parameters $\hat{\boldsymbol{\theta}}_\delta$ in $\boldsymbol{V}_{\delta s}(\boldsymbol{\theta}_\delta)$ $\boldsymbol{G}_\delta(\boldsymbol{\theta}_\delta)$ and $\boldsymbol{R}_{\delta s}(\boldsymbol{\theta}_\delta)$, by their REML estimates $\hat{\boldsymbol{\theta}}$. EBLUPs $\hat{\Delta}_\alpha$ are also functions of $\hat{\boldsymbol{\gamma}}$ and are obtained using Equation (7):

$$\hat{\Delta}_\alpha(\hat{\boldsymbol{\theta}}_\delta) = \boldsymbol{l}^t_{\delta\alpha}\hat{\boldsymbol{\beta}}_\delta(\hat{\boldsymbol{\theta}}_\delta) + \boldsymbol{m}^t_{\delta\alpha}\hat{\boldsymbol{v}}_\delta(\hat{\boldsymbol{\theta}}_\delta), \tag{7}$$

where $\hat{v}_\delta(\hat{\boldsymbol{\theta}}_\delta)$ equals:

$$\hat{v}_\delta(\hat{\boldsymbol{\theta}}_\delta) = \hat{\boldsymbol{G}}_\delta(\hat{\boldsymbol{\theta}}_\delta)\boldsymbol{Z}_{\delta s}^t\boldsymbol{V}_{\delta s}(\hat{\boldsymbol{\theta}}_\delta)^{-1}\{\boldsymbol{\delta}_s - \boldsymbol{X}_{\delta\alpha}\hat{\boldsymbol{\beta}}_\delta(\hat{\boldsymbol{\theta}}_\delta)\}. \tag{8}$$

It is important to note that for AOIs in unsampled stands (i.e., pixels in unsampled compartments or the unsampled stands themselves), estimation will be made assuming that the model fit for the sampled stands also holds for the unsampled stands. Under that assumption, $\boldsymbol{m}_{\delta\alpha}^t\hat{v}_\delta(\hat{\boldsymbol{\theta}}_\delta) = 0$ and $\hat{\Delta}_\alpha(\hat{\boldsymbol{\theta}}_\delta) = \boldsymbol{l}_{\delta\alpha}^t\hat{\boldsymbol{\beta}}_\delta(\hat{\boldsymbol{\theta}}_\delta)$ is a synthetic predictor.

### 2.5.4. MSE Estimators for the $\delta$-modeling Method

For all AOIs, the mean squared error of the EBLUP was estimated using the estimator provided by [32] and extended in [4] to account for the fact that AOIs can contain a small number of population units. This estimator is the sum of three components where the last one, $2g_{3,\alpha}(\hat{\boldsymbol{\theta}}_\delta)$, is a bias correction factor:

$$\hat{MSE}\{\hat{\Delta}_{\delta\alpha}(\hat{\boldsymbol{\theta}}_\delta)\} = g_{1\delta\alpha}(\hat{\boldsymbol{\theta}}_\delta) + g_{2\delta\alpha}(\hat{\boldsymbol{\theta}}_\delta) + 2g_{3,\alpha}(\hat{\boldsymbol{\theta}}_\delta), \tag{9}$$

The first term of (9) equals:

$$g_{1\delta\alpha}(\hat{\boldsymbol{\theta}}_\delta) = \boldsymbol{m}_{\delta\alpha}^t\{\hat{\boldsymbol{G}}(\hat{\boldsymbol{\theta}}_\delta) - \hat{\boldsymbol{G}}(\hat{\boldsymbol{\theta}}_\delta)\boldsymbol{Z}_{\delta s}^t\hat{\boldsymbol{V}}_{\delta\alpha}(\hat{\boldsymbol{\theta}}_\delta)^{-1}\boldsymbol{Z}_{\delta s}\hat{\boldsymbol{G}}(\hat{\boldsymbol{\theta}}_\delta)\}\boldsymbol{m}_{\delta\alpha} + \boldsymbol{q}_{\delta\alpha}^t\boldsymbol{R}_{\delta\alpha}(\hat{\boldsymbol{\theta}}_\delta)\boldsymbol{q}_{\delta\alpha}. \tag{10}$$

The second term of (9) is:

$$g_{2\alpha\delta}(\hat{\boldsymbol{\theta}}_\delta) = \boldsymbol{d}_{\delta\alpha}^t\{\boldsymbol{X}_{\delta s}^t\boldsymbol{V}_{\delta s}(\hat{\boldsymbol{\theta}}_\delta)^{-1}\boldsymbol{X}_{\delta s}\}^{-1}\boldsymbol{d}_{\delta\alpha} \tag{11}$$

with $\boldsymbol{d}_{\delta\alpha}^t = \boldsymbol{l}_{\delta\alpha}^t - \boldsymbol{m}_{\delta\alpha}^t\hat{\boldsymbol{G}}(\hat{\boldsymbol{\theta}}_\delta)\boldsymbol{Z}_{\delta s}^t\hat{\boldsymbol{V}}_{\delta s}(\hat{\boldsymbol{\theta}}_\delta)^{-1}\boldsymbol{X}_{\delta s}$. The term $g_{1\delta\alpha}(\hat{\boldsymbol{\theta}}_\delta)$ of $\hat{MSE}\{\hat{\Delta}_{\delta\alpha}(\hat{\boldsymbol{\theta}}_\delta)\}$ accounts for the uncertainty due to the estimation of the random effects while $g_{2\delta\alpha}(\hat{\boldsymbol{\theta}}_\delta)$ accounts for the uncertainty due to estimating $\boldsymbol{\beta}_\delta$.

For model (2), it is possible to compute a bias correction factor for the mean square error estimator, that accounts for the uncertainty due to estimating $\boldsymbol{\theta}_\delta$. This correction factor equals:

$$g_{3\delta\alpha}(\hat{\boldsymbol{\theta}}_\delta) = tr\left\{\left(\left.\frac{\partial \boldsymbol{b}_{\delta\alpha}^t}{\partial \boldsymbol{\theta}_\delta}\right|_{\hat{\theta}_y}\right)\boldsymbol{V}_{\delta s}(\hat{\boldsymbol{\theta}}_\delta)^{-1}\left(\left.\frac{\partial \boldsymbol{b}_{\delta\alpha}^t}{\partial \boldsymbol{\theta}_\delta}\right|_{\hat{\theta}_\delta}\right)^t\overline{\boldsymbol{V}}_{\delta s}(\hat{\boldsymbol{\theta}}_\delta)\right\}, \tag{12}$$

where, $\boldsymbol{b}_{\delta\alpha}^t = \boldsymbol{m}_{\delta\alpha}^t\boldsymbol{G}(\boldsymbol{\theta}_\delta)\boldsymbol{Z}_{\delta s}^t\boldsymbol{V}_{\delta s}(\boldsymbol{\theta}_\delta)^{-1}$ and $\overline{H}_{\delta s}(\hat{\boldsymbol{\theta}})$ is the Fisher information matrix for the fitted model. Explicit formulas for $g_{3\delta\alpha}(\hat{\boldsymbol{\theta}}_\delta)$ are provided [29] (pp. 179–180). This bias correction factor was used as a reference in comparisons with the $y$-modeling method.

All estimators of the mean square errors for AOIs in unsampled stands were made assuming that the model fitted for the sampled stands holds in the unsampled stands. Under this assumption, the leading term $g_{1\delta\alpha}(\hat{\boldsymbol{\theta}}_\delta)$, of $\hat{MSE}\{\hat{\Delta}_{\delta\alpha}(\hat{\boldsymbol{\theta}}_\delta)\}$ will be larger than if the stand containing the AOI was sampled. This occurs because the negative term $\boldsymbol{m}_{\delta\alpha}^t\hat{\boldsymbol{G}}(\hat{\boldsymbol{\theta}}_\delta)\boldsymbol{Z}_{\delta s}^t\hat{\boldsymbol{V}}_{\delta s}(\hat{\boldsymbol{\theta}}_\delta)^{-1}\boldsymbol{Z}_{\delta\alpha}\hat{\boldsymbol{G}}(\hat{\boldsymbol{\theta}}_\delta)\boldsymbol{m}_{\delta\alpha}$ makes the term $g_{1\delta\alpha}(\hat{\boldsymbol{\theta}})$ smaller as the stand sample size increases.

### 2.6. $y$-Modeling Method

#### 2.6.1. Model $y$-modeling Method

Models in the $y$-modeling method relate the forest structural variables in a population unit at different points in time with the auxiliary variables for that population unit. To indicate that these models directly consider the variables of interest, model parameters, stand-level random effects and model errors will include the subscript $y$. Auxiliary variables considered in the $y$-modeling method include the LiDAR auxiliary variables for 2009 and 2015, (i.e., Set 1 and Set 2, respectively) plus the incoming solar radiation and the factors structural diversity and presence of prescribed fires, Set 4 for

both 2009 and 2015. The modeling for the method started obtaining models for the variable of interest for 2009 and models for the variable of interest in 2015.

For given time $t$, the variable of interest in the $j^{th}$ population unit in the $i^{th}$ stand is expressed as:

$$y_{ijt} = x^t_{yijt}\beta_{yt} + u_{yit} + e_{yijt}, \tag{13}$$

where $x^t_{yijt}$ is a vector of auxiliary variables, specific for time $t$, in which the first element takes the value 1. The term $\beta_{yt}$ is a vector of time-specific coefficients with the first element representing the model intercept. The random components of model (13) are the stand-level random effects $u_{yit}$ and the model errors $e_{yijt}$. To account for heteroscedasticity, model errors $e_{yijt}$ were of the form $e_{yijt} = \varepsilon_{yijt}k^{\omega_{yt}}_{ijt}$ with $\varepsilon_{yijt} \sim N(0, \sigma^2_{y\varepsilon t})$; the term $k_{ijt}$, the predictor included in the model for time $t$, is most correlated to $y_t$, and $\omega_{yt}$ is a parameter to model the change in the error variance. The stand-level random effects $u_{yit}$ were assumed to be independently and identically distributed (i.i.d.) normal random variables $u_{yit} \sim N(0, \sigma^2_{yut})$ for all $i = 1, \ldots, D$, where $D$ is the total number of stands in the study area. Model errors were assumed independent of the stand-level random effects (i.e., $Cov(e_{yijt}, u_{ykt}) = 0$, for all $i$, $j$ and $k$). Finally, model errors were consider independent with $Cov(\varepsilon_{yijt}, \varepsilon_{yklt}) = 0$ if $i \neq k$ or $j \neq l$ for both $t = 2009$ and $t = 2015$. Models with spatially correlated errors were initially considered, but discarded for both years in the selection stage as no clear spatial correlation patterns were observed in the residuals. Auxiliary variables included in the model were selected following the same procedure used in the $\delta$-modeling method, using the best subsets selection procedure described in [27].

To account for expected correlations, models for 2009 and 2015 were combined into a single model where stand-level random effects and model errors for 2009 and 2015 were allowed to be time correlated. Then for the $j^{th}$ population unit in the $i^{th}$ stand the two-dimensional vector $y_{ij} = (y_{ij09}, y_{ij15})^t$ of variables of interest was related to the auxiliary variables through the following model:

$$y_{ij} = X_{ij}\beta_y + B_{ij}v_{yi} + e_{yij} \tag{14}$$

with:

$$X_{ij} = \begin{pmatrix} x^t_{yij09} & 0^t_{p15} \\ 0^t_{p09} & x^t_{yijt15} \end{pmatrix}, \beta_y = \begin{pmatrix} \beta_{y09} \\ \beta_{y15} \end{pmatrix}, B_{ij} = \begin{pmatrix} 1 & 1 & 0 \\ 1 & 0 & 1 \end{pmatrix}, v_{yi} = \begin{pmatrix} v_{yi} \\ v_{yi09} \\ v_{yi15} \end{pmatrix}, e_{yij} = \begin{pmatrix} e_{yij09} \\ e_{yij15} \end{pmatrix}, \tag{15}$$

where $0^t_{p2009}$ and $0^t_{p2015}$ are, respectively, row vectors of zeroes of dimensions equal to $x^t_{yij09}$ and $x^t_{yij2015}$.

As with the time-specific models, to account for heteroscedasticity in the combined model, model errors $e_{yijt}$ were of the form $e_{yijt} = \varepsilon_{yijt}k^{\omega_{yt}}_{ijt}$ with $\varepsilon_{yijt} \sim N(0, \sigma^2_{y\varepsilon t})$. The parameters $\omega_{yt}$ were updated when fitting the combined model. Spatial correlation of model errors was not found to be significant when considering each year separately, therefore, no spatial correlation patterns were considered in the combined model. The only source of correlation of model errors present in the combined model was temporal correlation. For a given location, the variables $\varepsilon_{yijtyij09} \sim N(0, \sigma^2_{y\varepsilon09})$ and $\varepsilon_{yij15} \sim N(0, \sigma^2_{y\varepsilon15})$ were allowed to be correlated random variables. The correlation between $\varepsilon_{yij2009}$ and $\varepsilon_{yij2015}$ is $\rho_\varepsilon$ and the variance-covariance matrix of $e_{yij}$ is:

$$Cov\begin{pmatrix} e_{yij09} \\ e_{yij15} \end{pmatrix} = R_{yij} = \begin{pmatrix} \sigma^2_{y\varepsilon09}k^{2\omega_{y09}}_{ij09} & \rho_\varepsilon\sigma_{09}k^{\omega_{y09}}_{ij09}\sigma_{09}k^{\omega_{y15}}_{ij15} \\ \rho_\varepsilon\sigma_{09}k^{\omega_{y09}}_{ij09}\sigma_{09}k^{\omega_{y15}}_{ij15} & \sigma^2_{yv}k^{2\omega_{y15}}_{ij15} \end{pmatrix}. \tag{16}$$

To model correlation between stand-level random effects, three random components $v_{yi}$, $v_{yi2009}$ and $v_{yi2015}$, independent of each other, were considered. These random components had distributions $v_{yi} \sim N(0, \sigma^2_{yv})$, $v_{yi09} \sim N(0, \sigma^2_{yv09})$ and $v_{yi15} \sim N(0, \sigma^2_{yv15})$. Stand-level random effect for a given point, at time $t$, $u_{yit}$, are the sum of a pure stand effect, independent of time $t$, $v_{yi}$, and a time-specific stand

random effect $v_{yi09}$ or $v_{yi15}$. The term $B_{ij}v_{yi} = u_{yi} = \begin{pmatrix} v_{yi} + v_{yi09} \\ v_{yi} + v_{yi15} \end{pmatrix} = \begin{pmatrix} u_{yi05} \\ u_{yi09} \end{pmatrix}$ represents these sums. The variance covariance matrix of $v_{yi}$ is diagonal, therefore, the variance covariance matrix of $u_{yi}$ is:

$$Cov\begin{pmatrix} u_{yi09} \\ u_{yi15} \end{pmatrix} = G_{yi} = \begin{pmatrix} \sigma_{yv}^2 + \sigma_{yv15}^2 & \sigma_{yv}^2 \\ \sigma_{yv}^2 & \sigma_{yv}^2 + \sigma_{yv15}^2 \end{pmatrix} \tag{17}$$

The fact that the random effect $v_{yi}$ is present for both 2009 and 2015 results in a positive correlation of the terms of $u_{yi}$, with a correlation coefficient $\rho_u = \frac{\sigma_{yv}^2}{(\sigma_{yv}^2 + \sigma_{yv09}^2)(\sigma_{yv}^2 + \sigma_{yv15}^2)}$. In a last step, models with a simpler structure of random effects were fitted and compared to the original models using a likelihood ratio test. Simplified models contained only random effects $v_{yi}$ that did not depend on time (i.e., models did not contain time-specific random effects $v_{yi09}$ and $v_{yi15}$). For simplified models $u_{yi09} = u_{yi15} = v_{yi}$.

For a generic set of population units $\xi$, the combined model can be expressed in matrix notation as:

$$y_\xi = X_{y\xi}\beta_y + Z_{y\xi}v_y + e_{y\xi}, \tag{18}$$

where $y_\xi$, $e_{y\xi}$, and $X_{y\xi}$, are obtained stacking the vectors $y_{ij}$, $e_{yij}$, or the matrices $X_{ij}$ of all units in $\xi$. As no spatial correlation patterns were found, the variance matrix of $e_{y\xi}$ is, $R_{y\xi} = diag_{i,j\in\xi}(R_{yij})$, a block diagonal matrix of dimension $2N_\xi x2N_\xi$ with $2x2$ blocks equal to $R_{yij}$. The vector of stand-level random effects $v = (v_{i1}^t, v_{i2}^t, \ldots, v_{iD}^t)^t$ and the matrix $Z_{y\xi}$ is an incidence matrix of dimension $2N_\xi xD$ for the simplified models and $2N_\xi x3D$ for the models with time-specific random effects. The variance covariance matrix of $y_\xi$ can be expressed as:

$$V_{y\xi}(\theta_y) = Z_{y\xi}G_y(\theta_y)Z_{y\xi}^t + R_{y\xi}(\theta_y). \tag{19}$$

In Equation (19), it is explicitly indicated that matrices $V_{y\xi}(\theta_y)$, $G_y(\theta_y)$ and $R_{y\xi}(\theta_y)$ depend on the vector of variance-covariance parameters $\theta_y = (\sigma_{yv}^2, \sigma_{yv09}^2, \sigma_{yv15}^2, \sigma_{y\varepsilon09}^2, \sigma_{y\varepsilon15}^2, \rho_\varepsilon)^t$. For the models with simplified random effects the vector of variance covariance parameters reduces to $\theta_y = (\sigma_{yv}^2, \sigma_{y\varepsilon09}^2, \sigma_{y\varepsilon15}^2, \rho_\varepsilon)^t$. Model (18) is a special case of linear mixed effect model with block diagonal covariance structure.

2.6.2. Target Parameter $y$-modeling Method

Under model (18) the target parameter (1) for a generic AOI, $\alpha$ is a linear combination of the form:

$$\Delta_\alpha = \frac{1}{6N_\alpha} \sum_{i=1}^{N_\alpha} (y_{i\alpha15} - y_{i\alpha09}) = l_{y\alpha}^t \beta_\delta + m_{y\alpha}^t u_y + q_{y\alpha}^t e_{y\alpha}, \tag{20}$$

where $l_{y\alpha}^t = q_{y\alpha}^t X_{y\alpha}$, $m_{y\alpha}^t = q_{y\alpha}^t Z_{y\alpha}$ and $q_{y\alpha}^t$ are vectors of known constants for the target AOI $\alpha$, with $q_{y\alpha}^t$ a vector of dimension $2N_\alpha$ where the $k^{th}$ element equals $\frac{(-1)^k}{6N_\alpha}$. It is important to remark that for models with a simplified structure of stand random effects, the target parameters do not depend on $u_y$. For these models, $y_{ij15} - y_{ij15} = (x_{yij15}^t\beta_{y15} - x_{yij09}^t\beta_{y09}) + (e_{yij15} - e_{yij09})$, and $u_{yi09} - u_{yi15} = v_{yi} - v_{yi} = 0$. For these type of models, one can expect significant gains in accuracy because it is not necessary to estimate random effects.

2.6.3. Estimator $y$-modeling Method, and Estimator of the MSE

Model (18) is a linear mixed effects model with block diagonal structure and $\Delta_\alpha$ a linear model parameter; thus, after [29] (pp. 108–110), the EBLUP $\hat{\Delta}_y(\hat{\theta}_y)$ of $\Delta_\alpha$ is:

$$\hat{\Delta}_{y\alpha}(\hat{\theta}_y) = l_{y\alpha}^t \hat{\beta}_y(\hat{\theta}_y) + m_{y\alpha}^t \hat{v}_y(\hat{\theta}_y), \tag{21}$$

where $\hat{\beta}_y(\hat{\theta}_y)$ equals:

$$\hat{\beta}_y(\hat{\theta}_y) = \{X_{ys}^t \hat{V}_{ys}(\hat{\theta}_y)^{-1} X_{ys}\}^{-1} X_{ys}^t \hat{V}_{ys}(\hat{\theta}_y)^{-1} y_s. \tag{22}$$

Matrices $\hat{V}_{ys}(\hat{\theta}_y)$, $\hat{G}_y(\hat{\theta}_y)$ and $\hat{R}_{ys}(\hat{\theta}_y)$ are obtained by replacing the estimated variance parameters $\theta_y$ in $V_{y\xi}(\theta_y)$, $G_y(\theta_y)$ and $R_{y\xi}(\theta_y)$, by their REML estimates $\hat{\theta}_y$. EBLUPs $\hat{\Delta}_\alpha$ are also functions of $\hat{\gamma}$ and are obtained using formula (7), where $\hat{v}_\delta(\hat{\theta}_\delta)$ equals:

$$\hat{v}_y(\hat{\theta}_y) = \hat{G}_y(\hat{\theta}_y) Z_{ys}^t V_{ys}(\hat{\theta}_y)^{-1} \{y_s - X_{y\alpha}\hat{\beta}_y(\hat{\theta}_y)\}. \tag{23}$$

As with the $\delta$-modeling method, estimates for AOIs in unsampled stands were made assuming that the model fit for the sampled stands also applied in the unsampled stands, which leads to $m_{y\alpha}^t \hat{v}_y(\hat{\theta}_y) = 0$ and $\hat{\Delta}_{y\alpha}(\hat{\theta}_y) = l_{y\alpha}^t \hat{\beta}_y(\hat{\theta}_y)$ is a synthetic predictor.

For all AOIs, the estimator of the mean square error of the EBLUP under the $y$-modeling method, $\hat{MSE}\{\hat{\Delta}_{y\alpha}(\hat{\theta}_y)\}$, is:

$$\hat{MSE}\{\hat{\Delta}_{y\alpha}(\hat{\theta}_y)\} = g_{1y\alpha}(\hat{\theta}_\delta) + g_{2y\alpha}(\hat{\theta}_y). \tag{24}$$

The terms $g_{1y\alpha}(\hat{\theta}_y)$ and $g_{2y\alpha}(\hat{\theta}_y)$ in (24) are analogous to those in (10) and (11) and have similar interpretation. To compute $g_{1y\alpha}(\hat{\theta}_y)$ and $g_{2y\alpha}(\hat{\theta}_y)$, matrices $\hat{G}_{\delta s}(\hat{\theta}_\delta)$, $\hat{R}_{\delta s}(\hat{\theta}_\delta)$, $\hat{V}_{\delta s}(\hat{\theta}_\delta)$ and $\hat{R}_{\delta\alpha}(\hat{\theta}_\delta)$ must be replaced by $\hat{G}_{ys}(\hat{\theta}_y)$, $\hat{R}_{ys}(\hat{\theta}_y)$, $\hat{V}_{ys}(\hat{\theta}_y)$ and $\hat{R}_{y\alpha}(\hat{\theta}_y)$. For the $y$-modeling method we did not compute the second-order correction factors.

### 2.7. Comparison of Methods

Methods were compared using three different criteria. First, we used general measures of accuracy providing the average error or uncertainty of prediction at the pixel-level (2.7.1); then, we compared methods using AOI-specific estimates and measures of uncertainty (2.7.2). Finally, we assessed the risk of generating biased predictions when using the $\delta$-modeling method and $y$-modeling method in unsampled stands (Section 2.7.3).

2.7.1. General Accuracy Assessment

To compare the $\delta$-modeling method and $y$-modeling method, a fist assessment was made using the cross-validated model mean squared error, *mRMSE*, and the model bias *mBias*:

$$mRMSE = \sqrt{\frac{\sum_{i,j\in s}^n (\delta_{ij} - \hat{\delta}_{ij})^2}{n}}, \tag{25}$$

$$mBias = \frac{\sum_{i,j\in s} (\delta_{ij} - \hat{\delta}_{ij})}{n}, \tag{26}$$

where $\delta_{ij}$ is the observed value of change for the $j^{th}$ plot included in the $i^{th}$ sampled stand and $\hat{\delta}_{ij}$ is the predicted value for that plot when model coefficients are obtained removing that plot from the training dataset. For the $y$-modeling method $\hat{\delta}_{ij}$ is obtained using the observed and fitted values of the variable of interest, as $\hat{\delta}_{ij} = \frac{1}{6}(\hat{y}_{ij15} - \hat{y}_{ij09})$ where $\hat{y}_{ij09}$ and $\hat{y}_{ij15}$ are the predictions of $y_{ij09}$ and $y_{ij15}$ are obtained fitting the corresponding $y$-model without the observations for plot $ij$. In addition, we computed *mRMSE* and *mBias* in terms relative to the average changes observed in the sampled plots. These quantities are denoted as $mRRMSE = mRMSE/\hat{\Delta}_f$ and $mRBias = mBias/\hat{\Delta}_f$ where $\hat{\Delta}_f$ is the mean of the changes observed in the field plots.

2.7.2. AOI-specific Comparisons.

For each of the considered areas of interest an estimate by each method (i.e., EBLUPs using either $\hat{\Delta}_{\delta\alpha}(\hat{\theta}_y)$ or $\hat{\Delta}_{y\alpha}(\hat{\theta}_y)$) and their corresponding mean square error estimators (i.e., $\hat{MSE}\{\hat{\Delta}_{\delta\alpha}(\hat{\theta}_y)\}$ or $\hat{MSE}\{\hat{\Delta}_{y\alpha}(\hat{\theta}_y)\}$) were available. First, for each AOI and method, we directly compared $\hat{\Delta}_{\delta\alpha}(\hat{\theta}_y)$ and

$\hat{\Delta}_{y\alpha}(\hat{\boldsymbol{\theta}}_y)$, and the square roots of $\hat{MSE}\{\hat{\Delta}_{\delta\alpha}(\hat{\boldsymbol{\theta}}_\delta)\}$ and $\hat{MSE}\{\hat{\Delta}_{y\alpha}(\hat{\boldsymbol{\theta}}_y)\}$. To simplify the notation, we will omit the subscript indicating the target AOI unless it is necessary and refer to $\hat{\Delta}_\delta$ as $\hat{\Delta}_y$. Similarly, after omitting the subscript $\alpha$, the AOI specific root mean square errors will be denoted as:

$$RMSE_\delta = \sqrt{\hat{MSE}\{\hat{\Delta}_{\delta\alpha}(\hat{\boldsymbol{\theta}}_\delta)\}}, \tag{27}$$

$$RMSE_y = \sqrt{\hat{MSE}\{\hat{\Delta}_{y\alpha}(\hat{\boldsymbol{\theta}}_y)\}}, \tag{28}$$

To perform an assessment relative to the predicted values the following coefficient of variation:

$$CV\{\hat{\Delta}_{\delta\alpha}(\hat{\boldsymbol{\theta}}_\delta)\} = \frac{\sqrt{\hat{MSE}\{\hat{\Delta}_{\delta\alpha}(\hat{\boldsymbol{\theta}}_\delta)\}}}{\hat{\Delta}_{\delta\alpha}(\hat{\boldsymbol{\theta}}_\delta)}, \tag{29}$$

$$CV\{\hat{\Delta}_{y\alpha}(\hat{\boldsymbol{\theta}}_y)\} = \frac{\sqrt{\hat{MSE}\{\hat{\Delta}_{y\alpha}(\hat{\boldsymbol{\theta}}_y)\}}}{\hat{\Delta}_{y\alpha}(\hat{\boldsymbol{\theta}}_y)}, \tag{30}$$

was computed for each AOI and method. Finally, for each AOI we compared $CV\{\hat{\Delta}_{\delta a}(\hat{\boldsymbol{\theta}}_y)\}$ to $CV\{\hat{\Delta}_{y\alpha}(\hat{\boldsymbol{\theta}}_y)\}$. To simplify the notation we will refer to these coefficients of variation as $CV_\delta$ and $CV_y$. Finally, for each sampled AOI we computed, using only the field information, the sample mean $\hat{\Delta}_{f\alpha}$ and its standard error:

$$SE_{f\alpha} = \sqrt{\frac{\sum_{i,j\in s}^{n_\alpha} \left(\delta_{ij} - \hat{\Delta}_{f,\alpha}\right)^2}{(n_\alpha - 1)n_\alpha}}, \tag{31}$$

and its coefficient of variation $CV_{f\alpha}$. In Equation (31), $n_\alpha$ is the number of field plots in the considered AOI and the sub-index $f$ is used to indicate that these quantities are calculated using only field data. Again, to simplify the notation we removed the sub-indexes $\alpha$ unless they were necessary. Finally, the sample mean and the coefficient of variation were then compared to their counterparts (7) and (29) and (21) and (30) obtained by the $\delta$-modeling method and $y$-modeling method, respectively.

### 2.7.3. Extrapolation to Thinned Stands

The fact that thinned stands were not represented in the sample of field plots raises the question of how applicable the models obtained are using either the $\delta$-modeling method or the $y$-modeling method to these stands. Applying the models to these stands involves a degree of extrapolation to a different population and a high risk of producing biased predictions. We assessed this risk by comparing the distributions of the LiDAR predictors included in the models for the $\delta$-modeling method and the $y$-modeling method for the sample of field plots to the distributions of the predictors in: (1) the sampled stands, (2) the unsampled and not thinned stands and (3) unsampled and thinned stands. Within each group (i.e., field plots, FP; sampled stands not thinned, SS; unsampled stands not thinned, UN; and unsampled stands subject to thinning, UT), we estimated density functions for each LiDAR predictor using a Gaussian kernel and a bandwidth determined using Silverman's rule [33]. Note that we considered two AOIs for the largest level of aggregation, the first one is SS and the other one, SA, is the union of SS and UN. We first considered each predictor separately and graphically compared their density functions. Predictors for 2009 and 2015 in the $y$-modeling method were considered separately. For each predictor and group, we computed the area of overlap, *AO*, with the density function for the field plots which takes value 0 if there is no overlap and value 1 if the distribution of the predictor in the considered group equals the distribution for the sample.

In addition to the area of overlap and aiming to consider all predictors in a given model at once, we calculated $\overline{NT2}$, the average of Mesgaran's novelty index *NT*2 for each model and group [34]. This quantity provides the average Mahalanobis distance from the pixels of the group to the mean of the sample of field plots, and it is expressed in terms relative to the maximum Mahalanobis distance

observed in the sample. Values of $\overline{NT2}$ above one indicate that on average pixels in a group are at a distance to the mean of the field plots larger than the distance from the extreme field observation to the mean of the field plots. We also calculated $\overline{NT2}_{mean}$, the average of $NT2$, but using the mean Mahalanobis distance as normalizing constant instead of the maximum. The reference value of one for $\overline{NT2}_{mean}$ indicates that the average Mahalanobis distance from pixels, to the mean of the field plots, is the same as the average of the Mahalanobis distances observed in the sample. Means and variance covariance matrices for computation of Mahalanobis distances are always estimated using the sample of field plots.

## 3. Results

### 3.1. Selected Models δ-modeling Method and y-modeling Method

Selected models for the δ-modeling method included auxiliary variables from Set 1 and Set 3. It was possible to find alternative models including fixed effects for the diversity treatments (i.e., predictors from Set 4) with similar values of *mRMSE* and *mRBias*; however, those models did not improve the model fit. From a practical point of view, models that only depend on the LiDAR variables but do not depend on the structural diversity treatments or the presence/absence of prescribed fires make the models more portable and applicable to stands without needing to know exactly which one of these treatments was applied. Considering that models using the structural diversity and presence of prescribed fires as predictors did not result in important gains in accuracy, we selected models that were not dependent on these treatments (Table 2).

**Table 2.** Summary models for the δ-modeling method. Model coefficients, standard errors of the model coefficients, variance parameters and general metrics for accuracy assessment are provided. Predictor acronyms are explained in Table A1. Coef is the value of the coefficient and Std.Error its corresponding standard error. V indicates volume, BA indicates basal area and B indicates biomass.

| Model | Predictor | Coef | Std. Error | $\hat{\sigma}^2_{\delta v}$ | $\hat{\sigma}^2_{\delta \varepsilon}$ | mRMSE | mRRMSE | mBias | mRBias |
|---|---|---|---|---|---|---|---|---|---|
| V(m³ ha⁻¹ year⁻¹) | Intercept | 1.16 | 0.31 | 0.50 | 10.53 | 3.47 | 241.99% | $-1.83 \times 10^{-4}$ | −0.01% |
| | δElev_P50$_{15\text{-}09}$ | 1.33 | 0.27 | | | | | | |
| | δPcFstAbv2$_{15\text{-}09}$ | 0.23 | 0.07 | | | | | | |
| BA(m² ha⁻¹ year⁻¹) | Intercept | 0.31 | 0.12 | 0.01 | 0.14 | 0.39 | 116.30% | $-8.2 \times 10^{-4}$ | −0.25% |
| | δPcAllAbv2$_{15\text{-}09}$ | 0.05 | 0.01 | | | | | | |
| | Elev_P75$_{09}$ | −0.03 | 0.01 | | | | | | |
| | PcAllAbv2$_{15\text{-}09}$ | 0.02 | <0.01 | | | | | | |
| B(Mg ha⁻¹ year⁻¹) | Intercept | 1.03 | 0.17 | 0.19 | 2.52 | 1.72 | 180.20% | $-1.09 \times 10^{-3}$ | −0.11% |
| | δElev_var$_{15\text{-}09}$ | 0.05 | 0.02 | | | | | | |
| | δElev_P50$_{15\text{-}09}$ | 1.03 | 0.20 | | | | | | |
| | δCRR$_{15\text{-}09}$ | −16.67 | 6.58 | | | | | | |

For the models in the δ-modeling method, the variance of random the effects, $\hat{\sigma}^2_{\delta v}$, was very small compared to the variance of the model errors, $\hat{\sigma}^2_{\delta \varepsilon}$, (Table 2). This indicated that, in this forest and for these variables, the use of synthetic estimators that do not account for the variability between stands should not cause a strong bias problem.

Models for the y-modeling method showed a pattern similar to that observed for the δ-modeling and only included predictors from Set 1 and Set 2 (Table 3). Errors showed non-constant variance patterns for all variables. The predictor most correlated with the variable of interest (i.e., the predictor used to model the error variance) was the same for 2009 and 2015. For V and B, the variance of model errors was a function of the square of the mean LiDAR elevation (Elev_mean²), and the exponents of the error variance function were very close to those obtained in [4,17,27] for V, and in [27] for B. For BA, variance of model errors was a function of the percentage of first returns above two meters (PcFstAbv2). Based on the results of the likelihood ratio tests, that for all variables resulted in p-values larger than 0.87, simplified models were selected and used for prediction.

**Table 3.** Summary models for the *y*-modeling method. Model coefficients, standard errors of the model coefficients, variance-covariance parameters and general metrics for accuracy assessment are provided. Covariate acronyms are explained in Table A1. Coef is the value of the coefficient and Std.Error its corresponding standard error. Coef is the value of the coefficient and Std.Error its corresponding standard error. V indicates volume, BA indicates basal area and B indicates biomass.

| Model | Year | Covariate | Coef | Std.Error | $\hat{\sigma}^2_{yu}$ | Kijt | $\omega_{yt}$ | $\hat{\sigma}^2_{yt\varepsilon}$ | $\rho_e$ | General Accuracy Metrics for Change Per Hectare and Year | | | |
|---|---|---|---|---|---|---|---|---|---|---|---|---|---|
| | | | | | | | | | | *mRMSE* | *mRRMSE* | *mBIAS* | *mRBias* |
| V(m³ ha⁻¹) | 2009 | Intercept | −19.09 | 10.36 | | | | | | | | | |
| | | $\text{Elev\_mean}^2_{09}$ | 2.52 | 0.23 | | $\text{Elev\_mean}^2_{09}$ | 0.64 | 3.00 | | | | | |
| | | $\text{PcFstAbv2}_{09}$ | 0.63 | 0.05 | 640.29 | | | | 0.85 | 3.76 | 262.62% | 0.13 | 9.24% |
| | 2015 | Intercept | 2.69 | 0.23 | | | | | | | | | |
| | | $\text{Elev\_mean}^2_{15}$ | 0.69 | 0.05 | | $\text{Elev\_mean}^2_{15}$ | 0.61 | 4.17 | | | | | |
| | | $\text{PcFstAbv2}_{15}$ | −26.30 | 11.10 | | | | | | | | | |
| BA(m² ha⁻¹) | 2009 | Intercept | −0.22 | 1.57 | | | | | | | | | |
| | | $\text{Elev\_P10}_{09}$ | −1.37 | 0.34 | | | | | | | | | |
| | | $\text{Elev\_P30}_{09}$ | 1.58 | 0.24 | | $\text{PcFstAbv2}_{09}$ | 0.48 | 0.81 | | | | | |
| | | $\text{PcFstAbv2}_{09}$ | 0.51 | 0.03 | 7.42 | | | | 0.85 | 0.47 | 138.06% | 0.01 | 1.53% |
| | 2015 | Intercept | −2.16 | 0.61 | | | | | | | | | |
| | | $\text{Elev\_P10}_{15}$ | 2.56 | 0.51 | | | | | | | | | |
| | | $\text{Elev\_P20}_{15}$ | 0.57 | 0.03 | | $\text{PcFstAbv2}_{15}$ | 0.45 | 1.12 | | | | | |
| | | $\text{PcFstAbv2}_{15}$ | −0.97 | 1.72 | | | | | | | | | |
| B(Mg ha⁻¹) | 2009 | Intercept | −11.86 | 5.19 | | | | | | | | | |
| | | $\text{Elev\_mean}^2_{09}$ | 1.19 | 0.11 | | $\text{Elev\_mean}^2_{09}$ | 0.71 | 0.39 | | | | | |
| | | $\text{PcFstAbv2}_{09}$ | 0.34 | 0.02 | 165.69 | | | | 0.85 | 1.94 | 203.69% | 0.08 | 8.60% |
| | 2015 | Intercept | 1.31 | 0.12 | | | | | | | | | |
| | | $\text{Elev\_mean}^2_{15}$ | 0.37 | 0.03 | | $\text{Elev\_mean}^2_{15}$ | 0.58 | 1.47 | | | | | |
| | | $\text{PcFstAbv2}_{15}$ | −14.15 | 5.76 | | | | | | | | | |

## 3.2. General Accuracy Assessment and Comparison of Methods

For all variables and modeling alternatives, values of *mBias* and *mRBias* were orders of magnitude smaller than *mRMSE* and *mRRMSE* (Tables 2 and 3). For all variables and methods, the percentages of explained variance for the change in V, BA and B were low. For the $\delta$-modeling method, models explained 34.38%, 31.37% and 39.04% of the variance of the change in V, BA and B, respectively. For the *y*-modeling method, models explained only 10.65% and 5.37% of V and B, respectively, while for BA the prediction using the *y*-modeling method was not better than the sample mean. In addition, $\delta$-models had values of *mRBias* lower than those obtained for the *y*-models. When instead of the change we considered the forest structural attributes with the *y*-modeling method, percentages of explained variance were 82.16% for V, 82.53% for BA and 82.93% for B. Considering only 2009, the percentage of explained variance for V, BA and B was 81.60%, 83.45%, and 82.84%, respectively. Considering only 2015, the percentage of explained variance for V, BA and B was 82.72%, 81.42% and 82.98%, respectively.

## 3.3. AOI-Specific Estimates

### 3.3.1. Entire Study Area

Estimates for the sampled stands and for the whole study area using either the $\delta$-modeling method or the *y*-modeling method were consistent with the estimates obtained using only the field information except for BA and B in SA. For the entire study area values of $RMSE_\delta$ tended to be smaller than $RMSE_y$. When considering the sampled stands, SS, approximate confidence intervals computed as $\hat{\Delta}_f \pm 2SE_f$ for the field estimates, and as $\hat{\Delta}_\delta \pm 2RMSE_\delta$ and $\hat{\Delta}_y \pm 2RMSE_y$ for each one of the LiDAR based methods, overlapped for all variables (Table 4) and contained estimates derived from other methods. Differences between the uncertainty of estimates obtained from LiDAR-based methods and the uncertainty of estimates obtained from field-based methods tend to be of small magnitude.

**Table 4.** Average increments of volume V, basal area BA and biomass B in the entire study area excluding the thinned stands (SA) and for the union of the sampled stands (SS). Estimates ($\hat{\Delta}$), root mean square errors ($RMSE$), coefficients of variation ($CV$) and confidence intervals ($CI$) obtained using the $\delta$-modeling method and the *y*-modeling method are compared to estimates ($\hat{\Delta}_f$), standard errors ($SE_f$) coefficients of variation ($CV_f$), and confidence intervals ($CI_f$) using only the field information.

| Variable | Area | $\delta$-modeling Method | | | | | *y*-modeling Method | | | | | Field Only Estimates | | | | |
|---|---|---|---|---|---|---|---|---|---|---|---|---|---|---|---|---|
| | | $\hat{\Delta}_\delta$ | $RMSE_\delta$ | $CV_\delta$ | $CI_\delta$ | | $\hat{\Delta}_y$ | $RMSE_y$ | $CV_y$ | $CI_y$ | | $\hat{\Delta}_f$ | $SE_f$ | $CV_f$ | $CI_f$ | |
| V(m$^3$ ha$^{-1}$ year$^{-1}$) | SS | 1.66 | 0.27 | 16.29% | 1.12 | 2.20 | 1.95 | 0.32 | 16.48% | 1.31 | 2.60 | 1.43 | 0.32 | 22.21% | 0.80 | 2.07 |
| | SA | 1.67 | 0.30 | 17.98% | 1.07 | 2.27 | 1.98 | 0.29 | 14.67% | 1.40 | 2.56 | | | | | |
| BA(m$^2$ ha$^{-1}$ year$^{-1}$) | SS | 0.36 | 0.03 | 8.68% | 0.30 | 0.42 | 0.37 | 0.04 | 9.93% | 0.30 | 0.45 | 0.34 | 0.04 | 10.87% | 0.26 | 0.41 |
| | SA | 0.42 | 0.04 | 8.41% | 0.35 | 0.49 | 0.44 | 0.04 | 9.61% | 0.35 | 0.52 | | | | | |
| B(Mg ha$^{-1}$ year$^{-1}$) | SS | 1.07 | 0.13 | 12.35% | 0.81 | 1.34 | 1.24 | 0.17 | 13.61% | 0.90 | 1.57 | 0.95 | 0.16 | 16.89% | 0.63 | 1.28 |
| | SA | 1.15 | 0.16 | 13.66% | 0.83 | 1.46 | 1.29 | 0.15 | 11.83% | 0.98 | 1.59 | | | | | |

### 3.3.2. Stands

Estimated change of V, BA and B in the sampled stands by both the $\delta$-modeling method and the *y*-modeling method agreed with their field-based counterparts in most stands (Figure 2). However, the width of the confidence intervals obtained using the $\delta$-modeling method tended to be larger than the confidence intervals of the estimates derived using the *y*-modeling method (Figure 2).

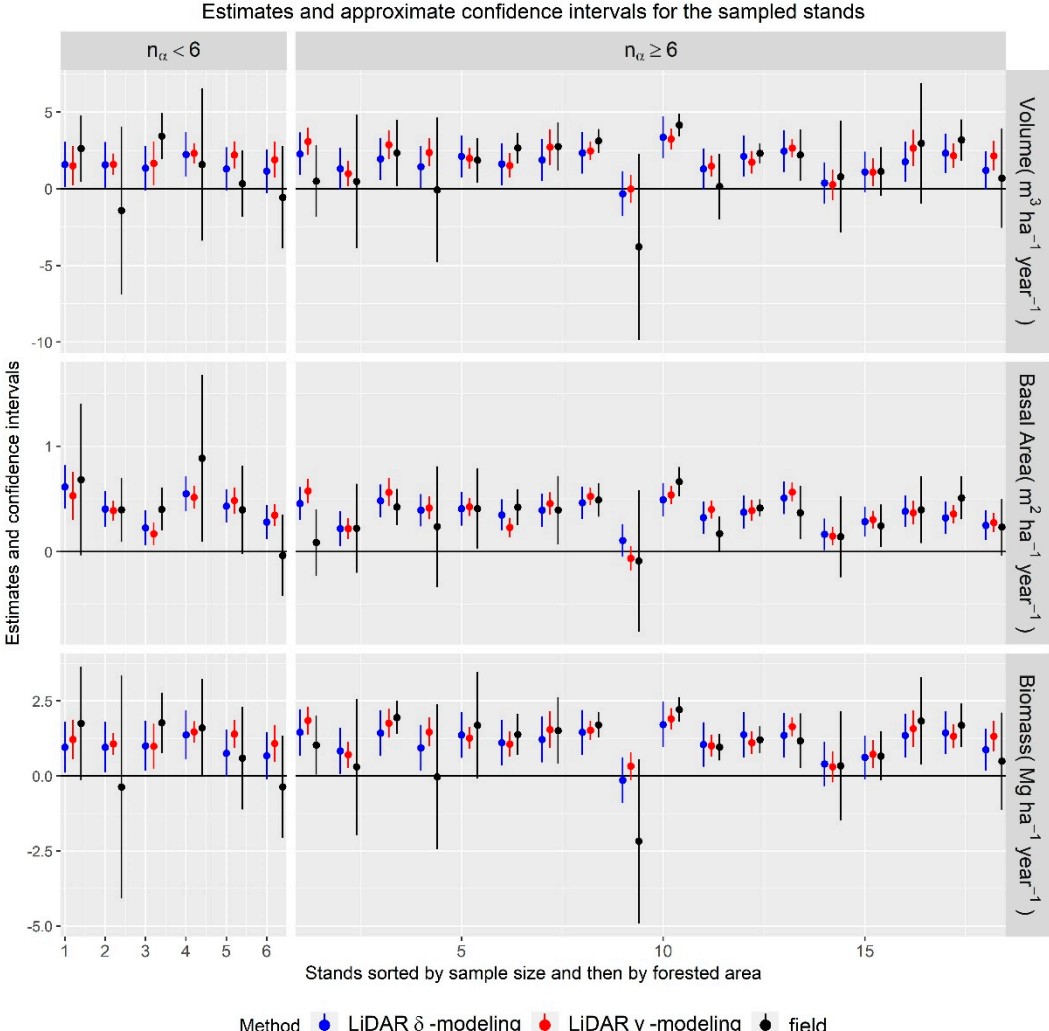

**Figure 2.** Estimates of V, BA and B change for the sampled stands of Blacks Mountains Experimental Forest. LiDAR-derived estimates using the $\delta$-modeling method are indicated by blue dots, LiDAR-derived estimates obtained using the $y$-modeling method are indicated with red dots and field-based estimates are indicated using black.

For unsampled stands, estimates and confidence intervals had larger variability in stands where the forested area was small (Figure 3). This variability cannot be avoided, and indicates that certain sources of errors cannot be compensated if the number of pixels that are aggregated is low. Finally, for both methods, values of $RMSE_\delta$ and $RMSE_y$ were in the range of 0.25 to 1 $m^3$ $ha^{-1}year^{-1}$ for V, of 0.02 to 0.15 $m^2$ $ha^{-1}year^{-1}$ for BA and of 0.10 to 0.80 Mg $ha^{-1}year^{-1}$ for B. However, for B and V, the $RMSE_y$ tended to be smaller than $RMSE_\delta$ while negligible differences between methods were observed for BA (Figures 3 and 4).

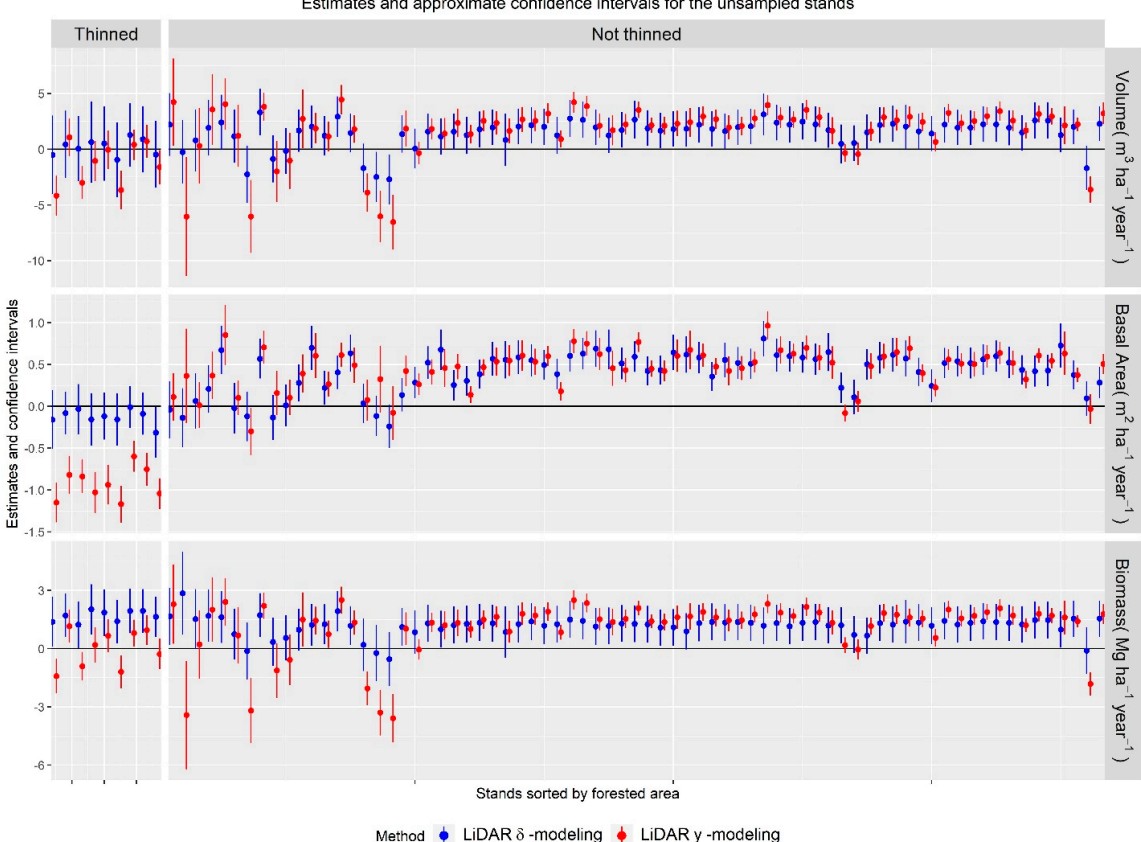

**Figure 3.** Estimates of V, BA and B change for the unsampled stands of Blacks Mountains Experimental Forest. LiDAR-derived estimates using the δ-modeling method are indicated by blue dots and LiDAR-derived estimates obtained using the *y*-modeling method are indicated with red dots. Thinned stands are to the left and non-thinned stands to the right.

For the thinned (and unsampled) stands, differences between δ-modeling method and the *y*-modeling method for BA were large and their confidence intervals did not overlap (Figure 3). For these stands, the estimates for BA using the δ-modeling method tended to indicate almost no changes in BA. Estimates for the thinned stands using the δ-modeling method provided inconsistent results indicating gains in B, and changes close to zero for V and BA. Certain inconsistencies were also observed for stands subject to thinning when using the *y*-modeling method where predictions of the change in V and B were positive for three and five stands respectively. These inconsistencies seem to derive from the fact that the distribution of predictors in Set 3 (i.e., changes in LiDAR predictors) in the thinned stands was rather different to the distribution of these predictors in the sample of field plots, in the sampled stands and in the unsampled and not thinned stands. For predictors of the *y*-modeling method, modeled differences between thinned stands and the remaining groups were of much smaller magnitude. Results for the analysis of the extrapolation risks are presented in detail in Section 3.4.

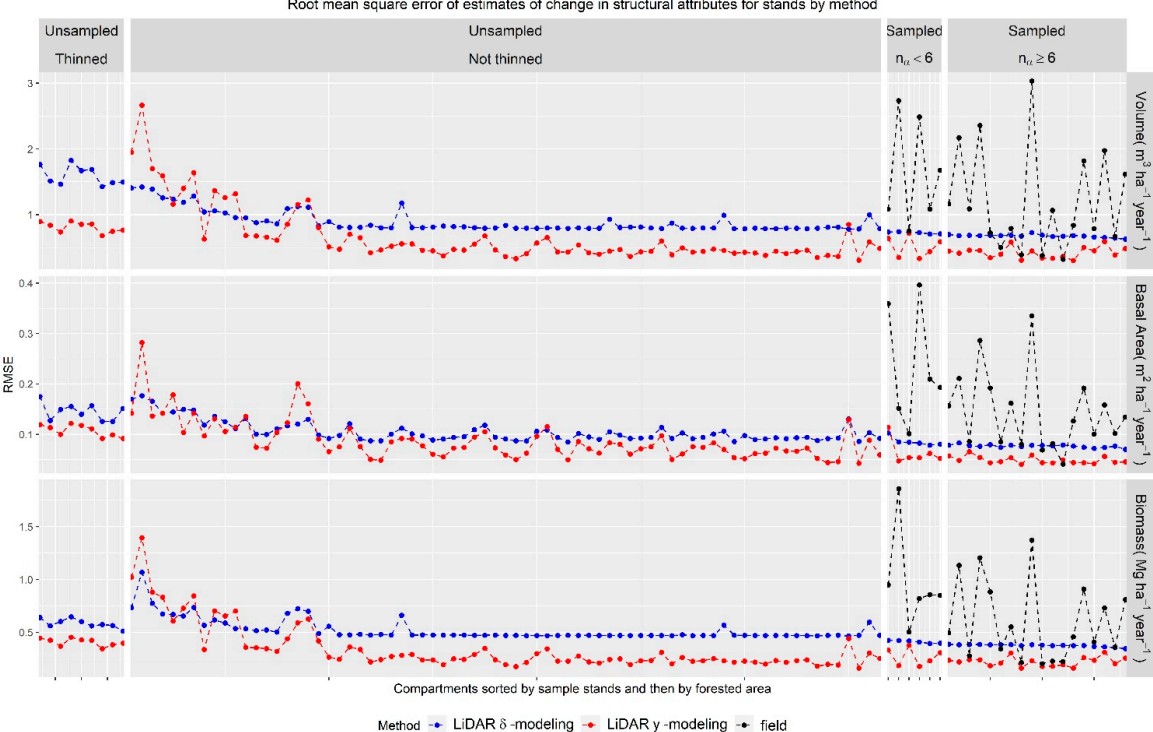

**Figure 4.** Values of $RMSE_\delta$ (blue), $RMSE_y$ (red) and $SE_f$ (black) for the stand-level estimates of V, BA and B.

### 3.3.3. Pixel-level

For both methods, inconsistencies observed at the stand level were observed at the pixel-level, especially the positive predictions of change obtained with the $\delta$-modeling method in the thinned stands (Figure A1). In addition, due to the low correlations of LiDAR predictors with the change in V, BA, and B, predictions at this level have large uncertainties. Mean and median values of $RMSE_\delta$ were 2.30 m$^3$ ha$^{-1}$ year$^{-1}$ and 3.30 m$^3$ ha$^{-1}$ year$^{-1}$ for V, 0.39 m$^2$ ha$^{-1}$ year$^{-1}$ and 0.38 m$^2$ ha$^{-1}$ year$^{-1}$ for BA, and 1.67 Mg ha$^{-1}$ year$^{-1}$ and 1.65 Mg ha$^{-1}$ year$^{-1}$ for B. Mean and median values of $RMSE_y$ were 2.49 m$^3$ ha$^{-1}$ year$^{-1}$ and 2.20 m$^3$ ha$^{-1}$ year$^{-1}$ for V, 0.48 m$^2$ ha$^{-1}$ year$^{-1}$ and 0.48 m$^2$ ha$^{-1}$ year$^{-1}$ for BA and 1.89 Mg ha$^{-1}$ year$^{-1}$ and 1.76 Mg ha$^{-1}$ year$^{-1}$ for B (Table 5 and Figure A1). Predictions from the $\delta$-modeling method tend to be smoother than predictions from the $y$-modeling method. For all variables, the proportion of pixel-level predictions using the $\delta$-modeling method within the range of values observed for the field plots, was always 99.84% or larger (Figure A2). Considering that these results were obtained in the presence of thinned stands and the relatively small fraction of the forest that was sampled, obtaining less than 0.16% of the predictions outside of the measurement range seems to be a clear sign of over smoothing (see Appendix B).

**Table 5.** Minimum (Min), 5th percentile (p05), mean, median, 95th percentile (p95) and maximum (Max) of $RMSE_\delta$ (27) and $RMSE_y$ (28) for the pixels of the study area.

| Variable | Method | Min | p05 | Mean | Median | p95 | Max |
|---|---|---|---|---|---|---|---|
| V(m$^3$ ha$^{-1}$ year$^{-1}$) | $\delta$-modeling method | 0.42 | 0.42 | 2.30 | 3.30 | 3.59 | 9.41 |
| | $y$-modeling method | 0.08 | 0.37 | 2.49 | 2.20 | 6.01 | 32.69 |
| BA(m$^2$ ha$^{-1}$ year$^{-1}$) | $\delta$-modeling method | 0.38 | 0.38 | 0.39 | 0.38 | 0.40 | 0.59 |
| | $y$-modeling method | 0.11 | 0.30 | 0.48 | 0.48 | 0.64 | 1.47 |
| B(Mg ha$^{-1}$ year$^{-1}$) | $\delta$-modeling method | 1.62 | 1.63 | 1.67 | 1.65 | 1.76 | 4.57 |
| | $y$-modeling method | 0.47 | 1.10 | 1.89 | 1.76 | 3.09 | 10.45 |

### 3.4. Extrapolation to Thinned Stands

Estimates of change in B for the thinned stands by both methods were clearly subject to bias problems. The predicted change in B for the total area subject to thinning for the period 2009–2015, using the $\delta$-modeling method was an increase in biomass of 40,469.22 Mg. The predicted change using the $y$-modeling method was a removal of B. However, the predicted removal for the period 2009–2015 was only 1750.29 Mg while the weighted extractions for the thinned stands were orders of magnitude larger. For BA both methods estimated extractions in BA, which is consistent with the fact that these stands were thinned. Estimated changes in BA using the $\delta$-modeling method for the thinned stands ranged from $-0.05$ m$^2$ ha$^{-1}$ to $-1.88$ m$^2$ ha$^{-1}$, which seems to be a very small change in basal area. Estimated changes in BA using the $y$-modeling method ranged from $-3.58$ m$^2$ ha$^{-1}$ to $-7.01$ m$^2$ ha$^{-1}$. An advantage of the $y$-modeling method is that it allows obtaining the values of the structural attributes at a given point in time. Using the $y$-model we estimated BA for the thinned stands for 2015. For those stands where thinning prescriptions dictated leaving a residual BA of 17.22 m$^2$ ha$^{-1}$ to 25.25 m$^2$ ha$^{-1}$, estimated BA for 2015 ranged from 19.87 m$^2$ ha$^{-1}$ to 26.22 m$^2$ ha$^{-1}$, which is in accordance with the thinning prescriptions. For the remaining area subject to thinning the estimated BA for 2015 was 17.64 m2 ha$^{-1}$, while the prescriptions dictated leaving a residual BA ranging from 6.89 m$^2$ ha$^{-1}$ to 13.77 m$^2$ ha$^{-1}$ in 75% of the area and leaving the remaining area untouched. In general, the estimated BA for 2015 are consistent with the prescriptions, which indicates that the $y$-modeling method produces reasonable estimates of BA when extrapolating to the thinned stands. In summary, for the estimation of changes, biases derived from extrapolation seemed to be of larger magnitude for the $\delta$-modeling method although they were also present for the $y$-modeling method.

The extrapolation indexes $\overline{NT2}$ and $\overline{NT2}_{mean}$ showed that predictions in thinned stands, involved a large amount of extrapolation when using the $\delta$-modeling method. For the $y$-modeling method, differences between thinned stands and stands not subject to thinning were of smaller magnitude (Figures 5 and A3). The inspection of the distribution of the LiDAR predictors in the field plots, the sampled and not thinned stands, unsampled and not thinned stands and the unsampled and thinned stands showed similar results for all variables, being the distributions of predictors from Set 3 (i.e., changes in LiDAR predictors for the period 2009–2015) very sensitive to the thinning operations (Figures 6, A4 and A5).

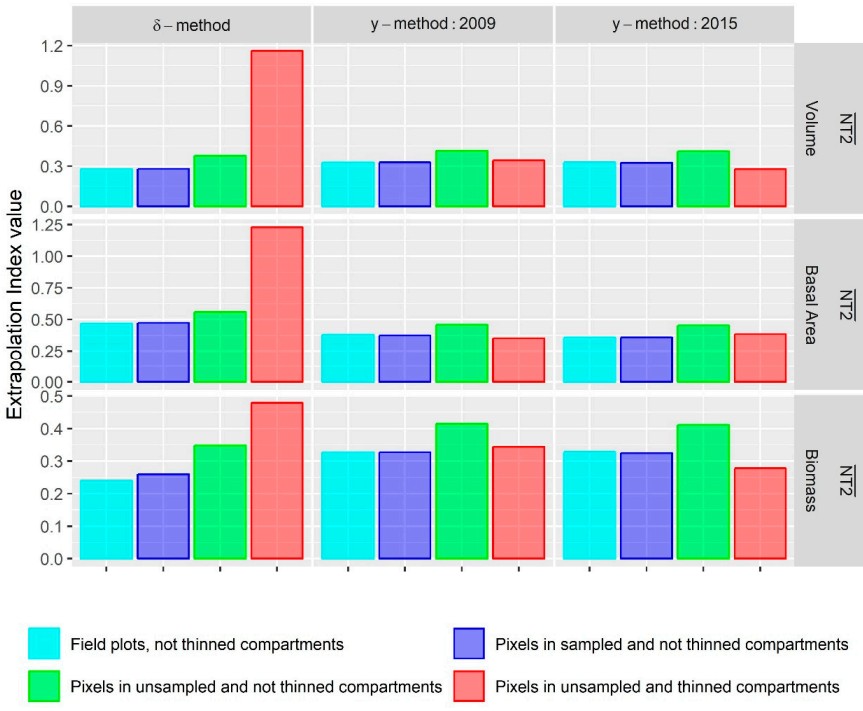

**Figure 5.** Indexes of extrapolation. Average of Mesgaran's novelty index [35], $\overline{NT2}$, for the sampled and not thinned stands (dark blue), unsampled stands not thinned (green) and unsampled and thinned stands (red). The value of this index for the field plots (light blue) provides the baseline value (i.e., the value observed for the sample of field plots).

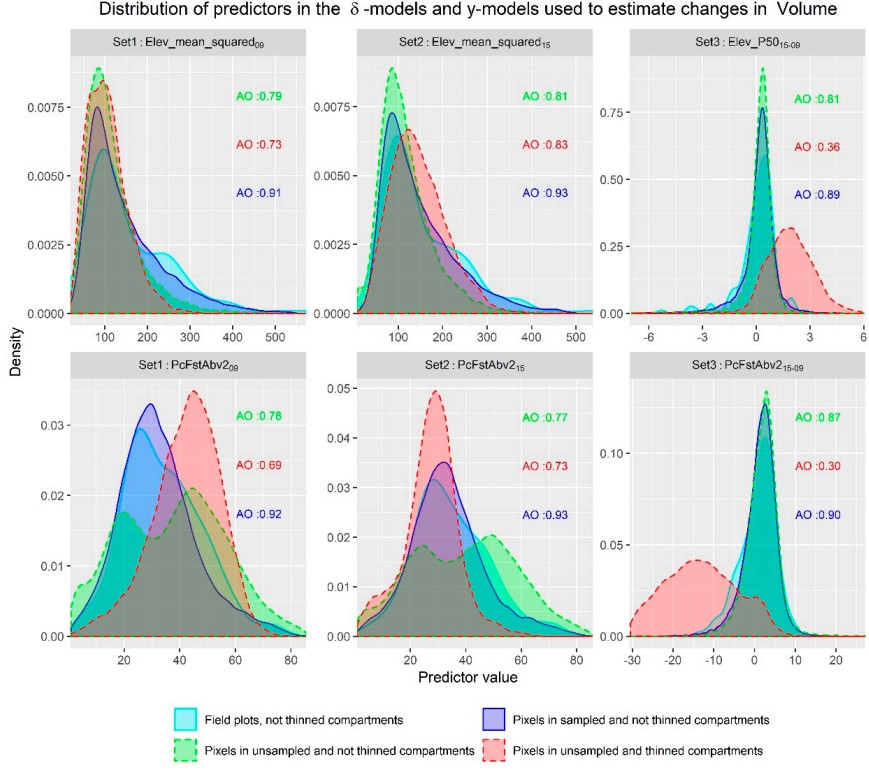

**Figure 6.** Comparison of density functions for the predictors in the models used to estimate changes in Volume using the $\delta$-modeling method and $y$-modeling method in field plots (light blue), sampled and not thinned stands (dark blue), unsampled and not thinned stands (light blue) and unsampled and thinned stands (red). For each group the area of overlap, AO, with the density function for the field plots (green) is provided for each predictor.

## 4. Discusion

*4.1. General Accuracy Assessment and Comparison of Methods.*

The smallest values of *mRMSE* were obtained using the δ-method, which is consistent with previous results reported by Poudel et al. [8] for V and B in coastal coniferous forest of Western Oregon and by Temesgen et al. [9] for B in spruce-dominated forest of Alaska (Tables 2 and 3). We observed, however, smaller differences between methods. Additionally, as observed in previous studies, (e.g., [8,21,35,36]) where LiDAR auxiliary variables showed a much stronger correlation with structural attributes at a given point in time than with their change.

Values of *mRMSE* for V were 3.47 m$^3$ ha$^{-1}$year$^{-1}$ when using the δ-method and 3.76 m$^3$ ha$^{-1}$ year$^{-1}$ when using the *y*-method. These values are slightly smaller than the *mRMSE* obtained by Poudel et al. [8] using the δ-method (4.74 m$^3$ ha$^{-1}$ year$^{-1}$) and two lidar acquisitions separated in time by five years. For B, *mRMSE* using the δ-method and the *y*-method were, respectively, 1.72 Mg ha$^{-1}$year$^{-1}$ and 1.94 Mg ha$^{-1}$ year$^{-1}$. These values were very close to those reported by Poudel et al. [8] using the δ-method (1.88 Mg ha$^{-1}$ year$^{-1}$) and worse than those reported by Temesgen et al. (1.25 Mg ha$^{-1}$year$^{-1}$ and 1.63 Mg ha$^{-1}$ year$^{-1}$), also using two LiDAR acquisitions separated in time by five years. Values of *RMSE* for BA were similar to those obtained by Næsset and Gobakken [20] in coniferous forest in Norway, using the *y*-method with log-transformed models and two LiDAR acquisitions that were two years apart from each other. In relative terms, for V and B, the values that we obtained for *mRRMSE* were considerably larger than those obtained by Poudel et al. [8]. These differences are due to the fact that observed growth rates in Poudel et al [8] are much higher than we observed at BMEF.

*4.2. AOI-Specific Estimates*

### 4.2.1. Entire Study Area

Most studies on estimation of change of structural variables using repeated LiDAR measurements have focused on analyzing indexes of model fit and reported only global measures of accuracy developed at a plot level. There is an important difference between the values of $RMSE_\delta$ and $RMSE_y$ and $mRMSE_\delta$ and $mRMSE_y$ being $mRMSE_\delta$ and $mRMSE_y$ an order of magnitude larger than $RMSE_\delta$ and $RMSE_y$. Model root mean square errors $mRMSE_\delta$ and $mRMSE_y$ provide an average measure of the errors that can occur when predicting a single pixel. For large areas, there will be some level of compensation of overpredicted and underpredicted pixels. Knowing how important that compensation is requires calculating AOI-specific root mean square errors. These AOI-specific measures cannot be directly derived from *mRMSE* because $RMSE_\delta$ and $RMSE_y$ consider factors such as the uncertainty in the estimation of the fixed and random effects that are not accounted for in $mRMSE_\delta$ or $mRMSE_y$. The effect of these factors in $RMSE_\delta$ and $RMSE_y$ can cause that the way two models with similar values of *mRMSE* rank based on this metric could change when attending to *RMSE*. But the most important consequence of the disconnect between $mRMSE_\delta$ and $mRMSE_y$ and AOI-specific measures of uncertainty, is that the former cannot be used as quality controls in LiDAR based inventories.

While numerous studies on estimation of changes using LiDAR rely on global measures of accuracy such as *mRMSE*, exceptions to this trend can be found in the literature [10,12,18–20]. The last four studies used model assisted techniques to derive either landscape or stratum level changes. Reported errors in those studies changed depending on the modeling techniques and study areas, but they all were of similar magnitude for changes in biomass per hectare and year (Table 4). Errors for the methods tested in this study were smaller than those reported by [10], where changes in live carbon stocks in Norway were estimated using generalized regression estimators (GREG). Differences with the errors reported in [10] for carbon, using the same 0.5 biomass to carbon conversion factor, were in the range of 0.12–0.09 Mg ha$^{-1}$ year$^{-1}$. These differences seem to be due to multiple factors such as differences between study areas, changes in live biomass versus changes in standing biomass,

time between LiDAR acquisitions and field plot sizes etc. Further investigation is needed to test if the model-based estimators studied here and the GREG estimators in [10] have a similar performance when used under the same conditions.

The study from Magnussen et al. [18] also included model based estimators using the $y$-modeling method. Reported errors were slightly larger than the ones observed here but at the same time smaller in terms relative to the observed mean change. An important result from the comparisons of [18] was the drastic improvement in model accuracy when developing stratum specific models (i.e., a set of model coefficients per stratum) as opposed to a global model for the whole study area. The mixed effect models used in this study can be used in combination with stratification if sample sizes are large. The introduction of stand level random effects allows for certain variability between AOIs that can be applied in situations where AOI sample sizes are limited.

### 4.2.2. Stands

One of the novelties of this study was the analysis of estimates for AOIs with small sample sizes to develop AOI specific models (i.e., stands). While at large scales both LiDAR-based and field-based estimates were very similar and had equivalent accuracies, at the stand-level, LiDAR based estimates, clearly had smaller errors than their field-based counterparts do. Qualitatively, this result for the change in V, BA and B is similar to the results obtained in [15,17] for the structural variables themselves and shows that the LiDAR auxiliary information allows for gains in efficiency when estimating changes in AOIs with small sample sizes. However, due to the low correlation of LiDAR and structural changes, values of $CV_\delta$ and $CV_y$ were oftentimes larger than 50%. These values of $CV$ are larger than those observed for structural variables in similar AOIs in previous studies [4,14,17]. While differences were not of large magnitude $RMSE_y$, tended to be smaller than $RMSE_\delta$. In addition, $RMSE_y$ had a larger variability because errors did not have constant variance. Finally, stand level estimates using the $\delta$-modeling method in the thinned and unsampled stands involved an important degree of extrapolation that can cause inconsistent estimates and severe biases, which indicates that the $\delta$-modeling method is more sensitive to extrapolating than the $y$-modeling method.

### 4.2.3. Pixel-level

For the most detailed level of disaggregation, the magnitude of the errors was very large. This is due to the low correlation between LiDAR auxiliary variables and the change in structural attributes. First-order and second-order texture indexes [37] are auxiliary variables with a promising potential for future research aiming to improve the prediction of structural changes. While for structural variables, maps at the pixel-level can provide a reliable reference about the forest structure; for growth and changes, pixel-level maps like the one in Figure A1 should be taken as mere approximations. They could be used to infer certain trends and patterns, but the high values of $RMSE_\delta$ and $RMSE_y$ show that estimates for a particular location made at the pixel scale can differ significantly with reality. These results clearly indicate that, predictions at such a fine scale are highly unreliable, and it is necessary either to perform some level of spatial aggregation or to increase the lapse between LiDAR acquisitions.

### 4.3. Advantages of Modeling Alternatives

In general, the $\delta$-modeling method was found to be a better alternative to estimate changes for the entire study area than the $y$-modeling method; however, the $y$-modeling method produced better results at the stand-level and also seemed to be advantageous to prevent problems related to extrapolation to values of the covariates outside of those included in model development.

The $\delta$-modeling method offers a faster model developments and fitting, and is significantly simpler than the modeling with the $y$-modeling method, as it is not necessary to consider differences between years and time correlations. The main disadvantage of this method is that it seems to be more prone to extrapolation errors. Predictors from Set 3 are sensitive to intense changes in the forest structure caused for example by thinning (see Figures 5, 6, A4 and A5). The inspection of predictors

of alternative models for V and B using this method revealed that inconsistencies of predictions in unthinned stands could be attenuated including more predictors from Set 1. The sensitivity to changes of predictors from Set 3 can be an advantage if all possible changes are correctly represented in the field sample. However, for relatively short periods of time between acquisitions and a low amount of forest operations, changes that are not very frequent in the landscape can be misrepresented or even not included in the sample. Thus, results for areas subject to those changes can be severely biased and inconsistent.

The more complex model development for the *y*-modeling method may be compensated by its ability to produce a richer set of outputs, by its apparently smaller risk of extrapolation and by its more accurate estimates for AOIs with small sample sizes (i.e., stands). In this study we analyzed the performance of the *y*-modeling method when estimating change, but estimates of V, BA and B for all the AOIs in 2009 and 2015 could have been readily obtained using this method. Results from our study also support the idea that the *y*-modeling method has advantages over the *δ*-modeling method in terms of protection against inconsistent extrapolations. The distributions of predictors from Set 1 and Set 2 in thinned stands were relatively similar to the distributions observed for the sample while the distributions of predictors in Set 3 used in the *δ*-modeling method, these distributions were rather different (see Figures 5, 6, A4 and A5). The greater similarity between thinned stands and the sample of field plots, for predictors from Set 1 and Set 2, indicates that the effect of thinning, in terms of auxiliary information, can be seen as transition from one situation in 2009 to another in 2015, and both seem to be represented in the field sample (e.g., Figure 5). If structures before and after the thinning (or other changes) are represented in the sample, the need for extrapolation will be limited. Within certain limits, if the sampling design covers all structures present at both points in time, even if there is a particular change from one structure to another that is not represented in the sample, predictions from the *y*-modeling method will not involve large extrapolations.

## 5. Conclusions

The four main conclusions obtained from this study include:

- The change of structural attributes and LiDAR auxiliary information are weakly correlated. This weak correlation seems to more evident in BMEF than in previous studies because of the slower growth in the study area and the relatively short lapse of time between LiDAR acquisitions, which indicates that for future studies in similar areas it might be necessary to increase the time lags between LiDAR flights.
- In general, the *δ*-modeling method was found to be a slightly more accurate alternative to obtain estimates of change for the whole study area; however, the *y*-modeling method was able to produce better estimates at the stand level. In addition, the *y*-modeling method method also seemed to be less prone to extrapolation problems. This indicates that field campaigns for the *δ*-modeling method have to be carefully designed while the *y*-modeling method might be less sensitive to certain bias problems.
- Despite the weak correlations with the changes in structural attributes, LiDAR auxiliary information allows obtaining estimates of growth for stands that improve over those derived using only field information.
- The large uncertainty observed for pixel-level predictions indicated that high-resolution maps of growth, generated using LiDAR auxiliary information in similar conditions, should be taken as approximated products.

**Author Contributions:** M.R. and B.W. developed the funding acquisition and data collection procedures. F.M. conceptualized and conducted the analyses and wrote the manuscript daft. M.R. and B.W. also participated in the conceptualization of the study. H.T., V.M., B.F. and A.H. provided significant input for the analyses and throughout the manuscript preparation.

**Funding:** This research received no external funding.

**Acknowledgments:** We would like to thank the USDA Forest Service, Region 5 and Lassen National Forest for assistance in project implementation, and the Guest Editor and two anonymous Reviewers for their constructive comments.

**Conflicts of Interest:** The authors declare no conflict of interest.

## Appendix A

**Table A1.** Sets of candidate predictors used in the study. Predictors included in the models to predict structural changes are highlighted with a boldface font. HiD, LoD and RNA represent the high diversity, low diversity and research natural areas respectively.

| Description Auxiliary Variables Sets 1, 2 and 3 | Acronym | | | Description Auxiliary Variables Set 4 | Acronym |
|---|---|---|---|---|---|
| | Set 1 Year: 2009 | Set 2 Year: 2015 | Set 3, Difference 2015-2009 | | Set 4 |
| Minimum, maximum, mean, mode, standard deviation, variance, coefficient of variation and interquartile range of the distribution of heights of the point cloud. | $Elev\_min_{09}$ $Elev\_max_{09}$ | $Elev\_min_{15}$ $Elev\_max_{15}$ | $\delta Elev\_min_{15-09}$ $\delta Elev\_max_{15-09}$ | Incoming solar radiation | Solar_radiation |
| | $Elev\_mean_{09}$ **$Elev\_mean^2_{09}$** $Elev\_mode_{09}$ | $Elev\_mean_{15}$ **$Elev\_mean^2_{15}$** $Elev\_mode_{15}$ | $\delta Elev\_mean_{15-09}$ $\delta Elev\_mean^2_{15-09}$ $\delta Elev\_mode_{15-09}$ | Structural diversity, factor with three levels HiD, LoD and RNA. Coded using two dummy variables. RNA reference level. | HiD |
| | $Elev\_stddv_{09}$ $Elev\_var_{09}$ | $Elev\_stddv_{15}$ $Elev\_var_{15}$ | $\delta Elev\_stddv_{15-09}$ **$\delta Elev\_var_{15-09}$** | | LoD |
| | $Elev\_CV_{09}$ | $Elev\_CV_{15}$ | $\delta Elev\_CV_{15-09}$ | Presence absence of prescribed fires. Coded using a dummy variable taking value 1 for stands where prescribed fires are applied and 0 otherwise. | Burned |
| | $Elev\_IQ_{09}$ $Elev\_AAD_{09}$ $Elev\_MADmed_{09}$ $Elev\_MADmod_{09}$ | $Elev\_IQ_{15}$ $Elev\_AAD_{15}$ $Elev\_MADmed_{15}$ $Elev\_MADmod_{15}$ | $\delta Elev\_IQ_{15-09}$ $\delta Elev\_AAD_{15-09}$ $\delta Elev\_MADmed_{15-09}$ $\delta Elev\_MADmod_{15-09}$ | | |
| Percentiles of the distribution of heights of the point cloud. | $Elev\_P01_{09}$ $Elev\_P05_{09}$ **$Elev\_P10_{09}$** $Elev\_P20_{09}$ **$Elev\_P30_{09}$** $Elev\_P40_{09}$ $Elev\_P50_{09}$ $Elev\_P60_{09}$ $Elev\_P70_{09}$ **$Elev\_P75_{09}$** $Elev\_P80_{09}$ $Elev\_P90_{09}$ $Elev\_P95_{09}$ $Elev\_P99_{09}$ | $Elev\_P01_{15}$ $Elev\_P05_{15}$ **$Elev\_P10_{15}$** **$Elev\_P20_{15}$** $Elev\_P30_{15}$ $Elev\_P40_{15}$ $Elev\_P50_{15}$ $Elev\_P60_{15}$ $Elev\_P70_{15}$ $Elev\_P75_{15}$ $Elev\_P80_{15}$ $Elev\_P90_{15}$ $Elev\_P95_{15}$ $Elev\_P99_{15}$ | $\delta Elev\_P01_{15-09}$ $\delta Elev\_P05_{15-09}$ $\delta Elev\_P10_{15-09}$ $\delta Elev\_P20_{15-09}$ $\delta Elev\_P30_{15-09}$ $\delta Elev\_P40_{15-09}$ **$\delta Elev\_P50_{15-09}$** $\delta Elev\_P60_{15-09}$ $\delta Elev\_P70_{15-09}$ $\delta Elev\_P75_{15-09}$ $\delta Elev\_P80_{15-09}$ $\delta Elev\_P90_{15-09}$ $\delta Elev\_P95_{15-09}$ $\delta Elev\_P99_{15-09}$ | | |
| Canopy relief ratio | $CRR_{09}$ | $CRR_{15}$ | **$\delta CRR_{15-09}$** | | |
| Percentage of first (Fst) and all (All) returns above 2 m | **$PcFstAbv2_{09}$** **$PcAllAbv2_{09}$** | **$PcFstAbv2_{15}$** $PcAllAbv2_{15}$ | **$\delta PcFstAbv2_{15-09}$** **$\delta PcAllAbv2_{15-09}$** | | |
| Ratio all returns above 2 m to first returns | $AllAbv2Fst_{09}$ | $AllAbv2Fst_{15}$ | $\delta AllAbv2Fst_{15-09}$ | | |
| Percentage of first returns above the mean and mode | $PcFstAbvMean_{09}$ $PcFstAbvMode_{09}$ | $PcFstAbvMean_{15}$ $PcFstAbvMode_{15}$ | $\delta PcFstAbvMean_{15-09}$ $\delta PcFstAbvMode_{15-09}$ | | |
| Percentage of all returns above the mean and mode | $PcAllAbvMean_{09}$ $PcAllAbvMode_{09}$ | $PcAllAbvMean_{15}$ $PcAllAbvMode_{15}$ | $\delta PcAllAbvMean_{15-09}$ $\delta PcAllAbvMode_{15-09}$ | | |
| Ratio of all returns above the mean and mode to number of first returns | $AllAbvMeanFst_{09}$ $AllAbvModeFst_{09}$ | $AllAbvMeanFst_{15}$ $AllAbvModeFst_{15}$ | $\delta AllAbvMeanFst_{15-09}$ $\delta AllAbvModeFst_{15-09}$ | | |
| Proportion of points in the height intervals [0,0.5), [0.5,1), [1,2), [2,4), [4,8) and [8,16) meters. | $Prop0\_05_{09}$ $Prop05\_1_{09}$ $Prop1\_2_{09}$ $Prop2\_4_{09}$ $Prop4\_8_{09}$ $Prop8\_16_{09}$ | $Prop0\_05_{15}$ $Prop05\_1_{15}$ $Prop1\_2_{15}$ $Prop2\_4_{15}$ $Prop4\_8_{15}$ $Prop8\_16_{15}$ | $\delta Prop0\_05_{15-09}$ $\delta Prop05\_1_{15-09}$ $\delta Prop1\_2_{15-09}$ $\delta Prop2\_4_{15-09}$ $\delta Prop4\_8_{15-09}$ $\delta Prop8\_16_{15-09}$ | | |

## Appendix B

Predictions from the $\delta$-modeling method tend to be smoother than predictions from the *y*-modeling method (Figure A1). For all variables, the proportion of pixel-level predictions using the $\delta$-modeling method within the range of values observed for the field plots, was always 99.84% or larger. Considering the presence of thinned stands and the relatively small fraction of the forest that is sampled. Obtaining less than 0.15% of the predictions outside of the measurement range seems to be a clear sign of over

smoothing. Predictions using the *y*-modeling method showed a greater variability, especially for BA, and the proportions of predictions inside the range of observed values, $P_y$, were 99.45% for V, 95.82% for BA and 99.29% for B. For BA, pixel-level predictions using the *y*-modeling method were oftentimes negative and of larger magnitude than the changes in BA observed for the plots. However, these pixels represent a small proportion of the total predictions (i.e., 4.02%), and a significant portion of them correspond to pixels in thinned stands. It is important to note that these comparisons of predicted values inform about how similar predictions are by the two analyzed methods and cannot be considered as indicators of accuracy or reliability. For all variables, pixel-level predictions by both methods were strongly correlated with Pearson's correlation coefficients of 0.92, 0.82 and 072 for V, BA and B, respectively (Figure A2). Finally, considering the unsampled and thinned stands, pixel-level predictions obtained by both methods showed the same inconsistencies observed at the stand-level especially for B using the *δ*-modeling method where only about 4%, of the predicted values were negative (i.e., removals of B). These inconsistencies are clearly due to extrapolations in the thinned stands and are analyzed in more detail in next section.

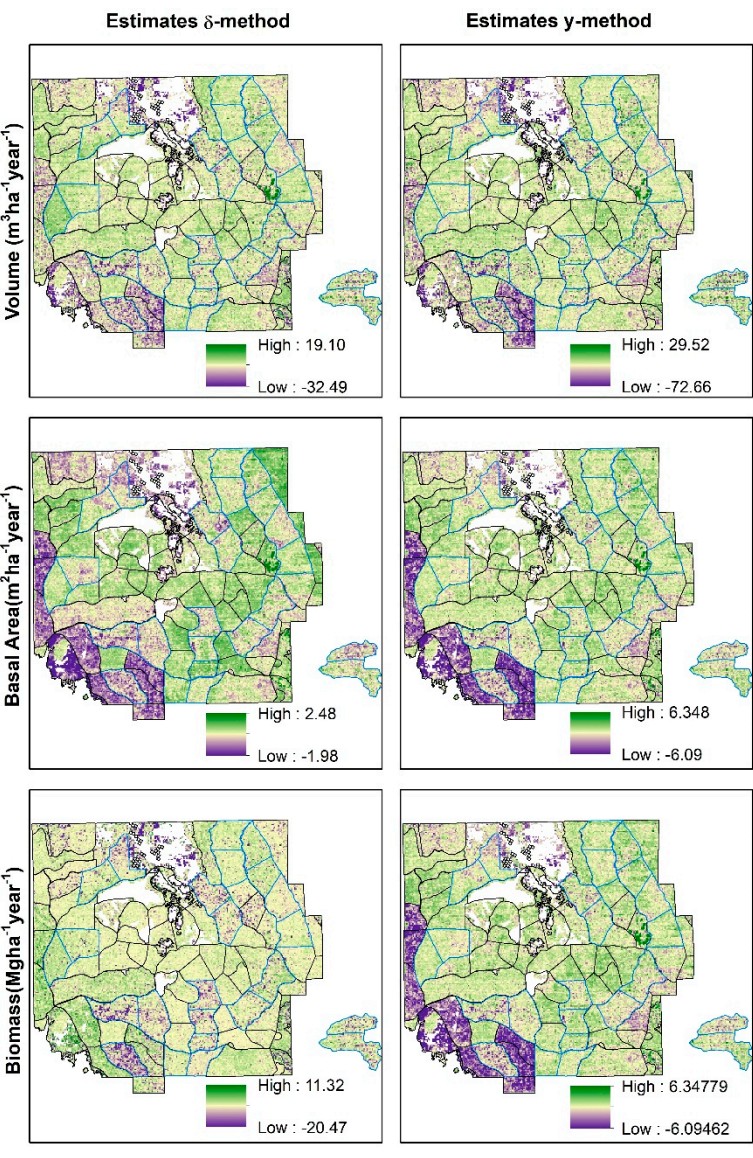

**Figure A1.** Maps of change in V, BA and B and corresponding pixel-level RMSE maps for the *δ*-modeling method.

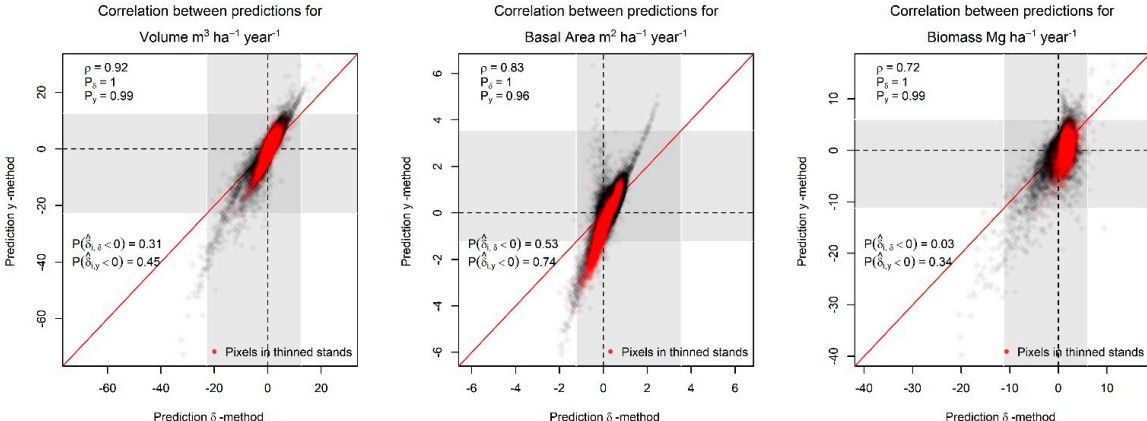

**Figure A2.** Comparison pixel-level predictions for V, BA and B using the δ-modeling method and *y*-modeling method predictions for the unsampled stands subject to thinnings are in red. The range of V, BA and B observed in the sample is indicated by the grey ribbons. The proportions, $P_\delta$ and $P_y$, of predictions within the range of values observed in the sample, and the correlation between predictions from both methods are indicated in the upper left corner. The proportion of pixels in the thinned stands where the δ-modeling method and *y*-modeling method predict losses (i.e., $P(\hat{\delta}_{i,\delta} < 0)$ and $P(\hat{\delta}_{i,y} < 0)$) are indicated on the lower left quadrant of the figure.

**Appendix C**

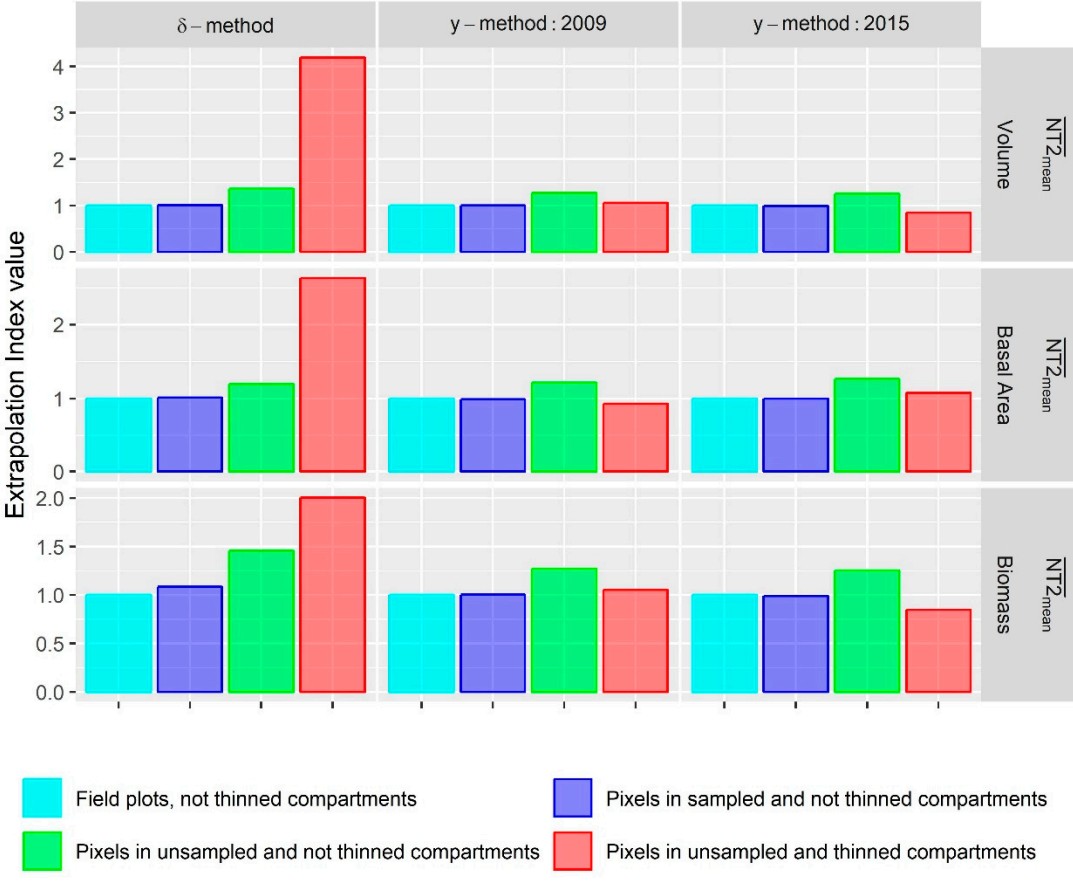

**Figure A3.** Indexes of extrapolation. Average of Mesgaran's novelty index relative to the mean, $\overline{NT2}_{mean}$, for the sampled and not thinned stands (dark blue), unsampled stands not thinned (green) and unsampled and thinned stands (red). The value of this index for the field plots (light blue) provides the baseline value (i.e., the value observed for the sample of field plots).

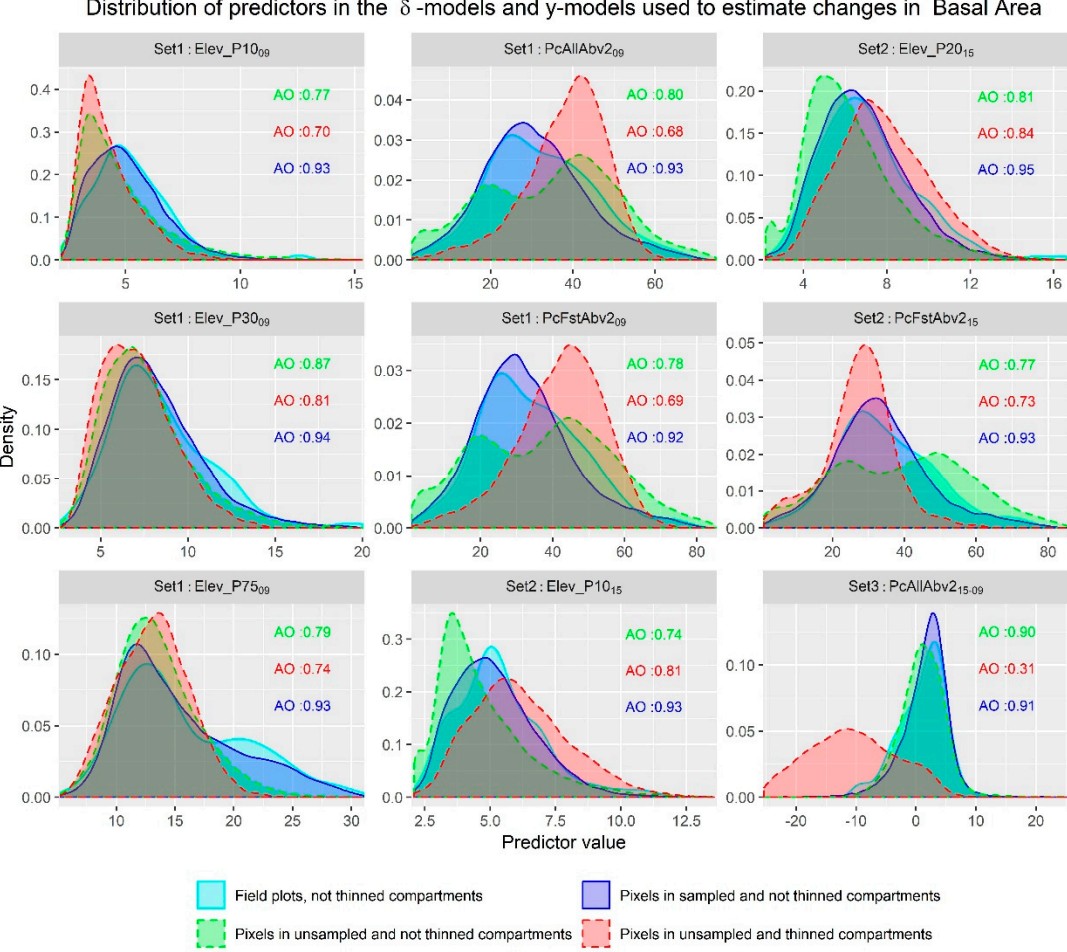

**Figure A4.** Comparison of density functions for the predictors in the models used to estimate changes in Basal Area using the $\delta$-modeling method and $y$-modeling method in field plots (light blue), sampled and not thinned stands (dark blue), unsampled and not thinned stands (light blue) and unsampled and thinned stands (red). For each group the area of overlap, AO, with the density function for the field plots (green) is provided for each predictor.

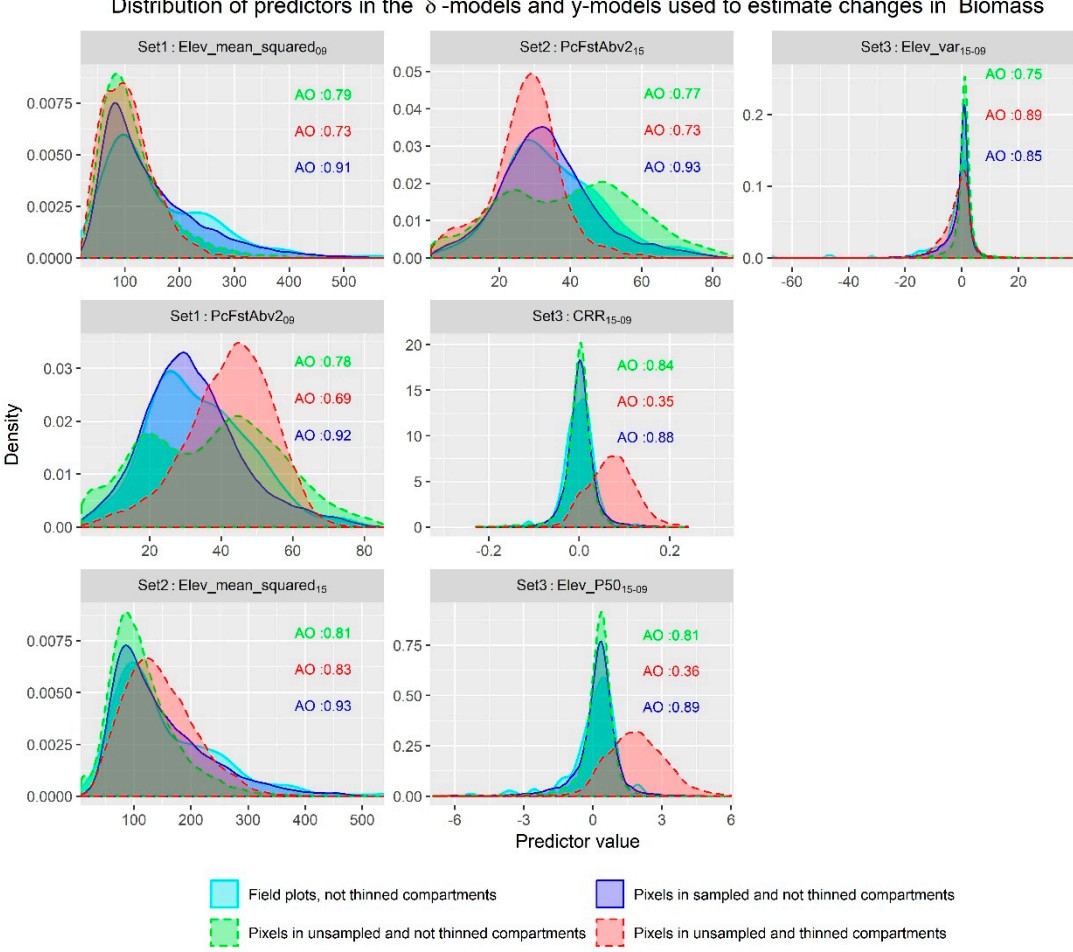

**Figure A5.** Comparison of density functions for the predictors in the models used to estimate changes in Biomass using the δ-modeling method and *y*-modeling method in field plots (light blue), sampled and not thinned stands (dark blue), unsampled and not thinned stands (light blue) and unsampled and thinned stands (red). For each group the area of overlap, AO, with the density function for the field plots (green) is provided for each predictor.

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
