# Peer review of "Estimation of Changes of Forest Structural Attributes at Three Different Spatial Aggregation Levels in Northern California using Multitemporal LiDAR"

_remotesensing, doi:10.3390/rs11080923_

Round 1

Reviewer 1 Report

Dear Authors; It was a great pleasure to read and review your paper. I can easily say that your manuscript was meticulously prepared and the findings are important. Even so, I have some questions and concerns given below;

1) In abstracts: the abstract was separated as Research Highlights, Background and Objectives, and Materials and Methods. I have not encountered this structure afore in the Remote Sensing. Please check it!

2) In introduction; Motivation is little weak and the originality of the study is hidden. Lidar-based estimation of forest attributes and the changes of stand structure have been extensively worked in literature. Authors should convince readers about the necessity and importance of the study.

3) It is expected that the changes 2009-2015 in structural attributes a natural forest stand (except rapid grown tree species such as poplar, corsican pine) are too small to be detected by LiDAR height metrics. I would see a large time period such as 10-15 years.

4) Why did you exclude the stands subject to thinning during the period between the two available LiDAR flights? The structural changes in these stands were able to be detected using LiDAR metrics? I think that thinned stands might be included. Make a sense?

5) Did you ensure that forest owner did not cut any trees? The measurements for two dates (2009-2016) were made by same team?

6) Which equation was used for calculating “solar radiation”? You mean a radiation index? RI=(1-cos((π/180)(aspect-30)))/2. Please add the equation.

7) Authors used the fitting statistics such as root mean squared error, standard error, bias for comparing the models. I recommend the authors to use an independent or cross-validation test for the evaluation of the predictive power and stability of the models. At least, if possible, a one-leave-out cross validation test would be good for the paper? Or explain why did not use a validation test?

8) Did you check the linear model assumptions (e.g. the normality of residuals?)

9) In table 3: Using different predictor variables (e.g. For Basal area, Elev_P30 for 2009 LiDAR vs Elev_P20 for 2015 LiDAR) for building models might be somewhat unfavorable for comparison? Is not it?

10) Point sizes of labels in the figures such as Fig6, fig7, fig8 should be increased.

11) Broadly speaking, extra features such as image texture extracted from the image are increasingly applied to improve change detection accuracy. /kernel assessed across the image. So, the addition of image texture (which normally refers to the measurements of the spatial variability of neighboring pixels within a moving window) derived from LiDAR CHMs might be more effective than using LiDAR metrics alone for change detection. I do not suggest a further process for this paper. However, a short discussion about image texture derived from CHM should be added to the discussion section. (A useful source is “Modelling tree size diversity from airborne laser scanning using canopy height models with image texture measures”)

12) According to my opinion, the paper is a little bit long for readers. It would be nice if you could shorten it.

Author Response

Please find attached our response to the provided comments.

Reviewer 2 Report

This paper presents a comparison of the so-called delta method (i.e. predicting the change using a model estimated for change directly) or y-method (i.e. estimating the state in the beginning and end of the period of interest and calculating the difference). I was quite skeptical in the beginning, as the delta method should by definition be optimal (i.e. produce minimum variance) for the change. That is, the direct change model is optimized for the change, but the state models are optimized for state, not change. So, when a model is optimal, it cannot be improved by using a more complicated model, and the y-model is just that, a more complicated model.

This can be seen if we define y1 = f1(x1,b1) and y2 = f2(x2,b2). The best estimates for b1 and b2 can be obtained with an OLS (in a simple case). We calculate the change as c = y2 -y1 = f2(x2,b2)-f1(x1,b1). Whatever the model forms f1 and f2, we can always estimate a new model g (x1,x2,c1,c2) that has the same shape as f2-f1, but where the parameters c1 and c2 are optimal for the change. Then, this model, which is again a direct model, has to be better than the f2-f1 model. So, in fact the y-method is also a direct method, just not optimal one. (In this I assume that the change model can also include Lidar variables from either time point as such, not only their differences. Besides, I think using just the differences is an unnecessary restriction that should be avoided).

So, the results obtained in this study, with the direct method more accurate than the y-method, were really not a surprise, and should not come as a surprise for the authors either.  For a discussion on the topic, see e.g. Massey, A., and Mandallaz, D., 2015. Design-based regression estimation of net change for forest inventories. Can. J. For. Res. 45: 17751784. Section 3.2.3. I think that reference should also be included into this paper.

However, having said that, I got interested when I saw that the authors have done the y-method models using simultaneous models and taking the correlation in time into account. They noted that they had clearly better results that the previous papers, and also noted that this might be related to the fact that they accounted for the model correlations (i.e. used simultaneous models). I agree. It seems that if the model is optimized for both the state 1 and 2 at the same time, it is semi-optimized also for the change, which makes the results better than usually with the y-method. I think that was a real good idea, and to me this is the most important result of this paper, not the change in the results due to aggregating.

I think that this approach could be taken even further, by including, for instance, the current state y2 and the change c into a simultaneous model. Or maybe even all, i.e. y1, y2 and c. That should efficiently combine the optimal parameter estimates (and possibly a more stable model, which might be the only true benefit of the y-method, also confirmed by the extrapolation results the authors obtained section 3.4) into the same package. Such an approach might also reduce the potential for bias now observed in the y-method (line 589).

When calculating results at aggregated level, and assuming the errors as uncorrelated, the error variance of mean of m observations would be the residual error variance divided by m (now excluding the errors in the parameter estimates that do not diminish). Since the residual variance of the direct method is (by definition) smaller than that of the y-method, this relationship should also be evident in the aggregated level, but for the results of small stands this does not seem to hold (line 621). However, as the errors were assumed to be correlated, and an EBLUP prediction was used, the within-AOI correlations should explain the results that the y-method had narrower confidence intervals than the direct method. Otherwise it is just counter-intuitive. The authors address this by saying that certain sources of errors cannot be compensated if the number of pixels that are aggregated is low. However, this should apply to both methods and therefore is not a good explanation. The higher the within-AOI correlations, the smaller the proportion of errors that can be compensated, and the larger is the AOI (or stand) level bias in the results. Because the correlations are the most likely explanation for the results, the authors should show the differences in the estimated correlations (Appendix?) and make the role of correlations more clear in how they describe the results.

The other possible explanation is the biased model. For the change predictions for the thinned stands, the results obtained with direct method were poor, and that can be understood as the modelling data was not representative enough for them. So, I agree with the authors that the y-method may be safer against the representativeness problems. I also agree that is it wise to add explanatory variables from the structural variables themselves, not just the differences. (This relates also with what I commented about making a simultaneous model for the state and the change and the y-method being also a direct method when presented differently).

Overall, the results section was quite long, and contained a lot of details and numbers. This makes it quite tedious read. I would like the authors to consider if all these numbers in the text are necessary, and drop numbers that are presented or can be presented in tables and figures, and just point out the most important ones.

Details:

Abstract has two background and objectives parts, and the first of them is included already in the highlights. It should be dropped.

Line 173: The authors sampled 26 stands and then 9 thinned stands were separated, but how did the authors end up with 15 units selected for sampling? I could not find a way to end up with 15, even though I tried assumptions like 26-9=17 and 151/26=5.8 so on. So, please explain where the 15 comes from.

Line 190: Please explain what edge effects? Were these plots outliers? Which kind of outliers?

213: I thought the lidar was from 2009?

251: Where does the 6 come to the formula of K?

399: Why can’t the time-specific stand random effect be just time-effect that is nested within a stand?

589: The authors say that the y-method “may suffer from bias problems”. The change in volume is estimated to be 1.08 from the field data, and 1.08+2*0.30 = 1.68, and the mean estimated from y-method is 1.60? Is this a correct interpretation? Please be more specific here about what you mean.

Line 607: According to mRMSE direct and mRMSE y the direct method is always superior. It should be obvious from my previous comments why it is and will always be superior.

Line 608: The authors refer to mRMSE here, but Table 4 does now show them. Do you mean RMSE?

Figures 2 and 3 are really very hard to read and understand. A figure with whiskers for each stand rather than lines such as used now, could improve the comprehensibility of the results. Figure 4 was much easier to understand.

Author Response

Please find attached my response to the comments provided.
